# ON-DEVICE COLLABORATIVE LANGUAGE MODELING VIA A MIXTURE OF GENERALISTS AND SPECIALISTS

## ABSTRACT

On-device LLMs have gained increasing attention for their ability to enhance privacy and provide a personalized user experience. To facilitate learning with private and scarce local data, federated learning has become a standard approach, though it introduces challenges related to system and data heterogeneity among end users. As a solution, we propose a novel **Co**llaborative learning approach with a **Mi**xture of **G**eneralists and **S**pecialists (CoMiGS), being the first to effectively address both. Our approach distinguishes generalists and specialists by aggregating certain experts across end users while keeping others localized to specialize in user-specific datasets. A key innovation of our method is the bi-level optimization formulation of the Mixture-of-Experts learning objective, where the router is updated using a separate validation set that represents the target distribution. CoMiGS effectively balances collaboration and personalization, as demonstrated by its superior performance in scenarios with high data heterogeneity across multiple datasets. By design, our approach accommodates users' varying computational resources through different numbers of specialists. By decoupling resource abundance from data quantity, CoMiGS remains robust against overfitting—due to the generalists' regularizing effect—while adapting to local data through specialist expertise.

## 1 INTRODUCTION

Large Language Models (LLMs) have been showing great success serving as foundation models, evidenced by their capability to understand a wide range of tasks, such as ChatGPT (OpenAI, 2023), Claude (Anthropic, 2023), Gemini (DeepMind, 2023) and etc. However, cloud-based inference introduces significant delays for end users, and it often fails to meet their personalized needs (Ding et al., 2024; Iyengar & Adusumilli, 2024). Recently, there has been growing interest in deploying LLMs on edge devices, which offer benefits like lower latency, data localization, and more personalized user experiences (Xu et al., 2024). For instance, Apple (2024) recently launched on-device foundation models as part of its personal intelligence system.

On-device LLMs present challenges such as limited and variable computational resources, scarce and heterogeneous local data, and privacy concerns related to data sharing (Peng et al., 2024; Wagner et al., 2024). Fine-tuning is typically performed on-device to quickly adapt to users' individual needs. While data sharing is a common solution to address local data scarcity, on-device data is often privacy-sensitive and must remain on the device. To overcome this, federated learning has been proposed as a method for enabling collaborative learning while preserving user privacy, allowing end users to collaborate by sharing model parameters (Chen et al., 2023; Zhang et al., 2023).

Federated fine-tuning of LLMs is predominantly done through Low-Rank Adaptation (LoRA, Hu et al. (2021)) due to its lightweight nature so that the communication costs can be largely mitigated. Yet end devices may have different capacities, resulting in different LoRA ranks or different numbers of LoRA modules allowed on devices. Previous works have proposed various techniques for aggregating LoRA modules of different ranks (Cho et al., 2023; Bai et al., 2024). However, in both works, the devices are only equipped with shared knowledge, which makes the methods unsuitable when there is data heterogeneity across users. In such cases, a more personalized solution is needed.

End users' local data distributions can exhibit significant statistical heterogeneity. For instance, mobile device users may have distinct linguistic habits, topic preferences, or language usage patterns, leading to widely varying word distributions. As a result, personalized solutions are necessary. Wagner et al.

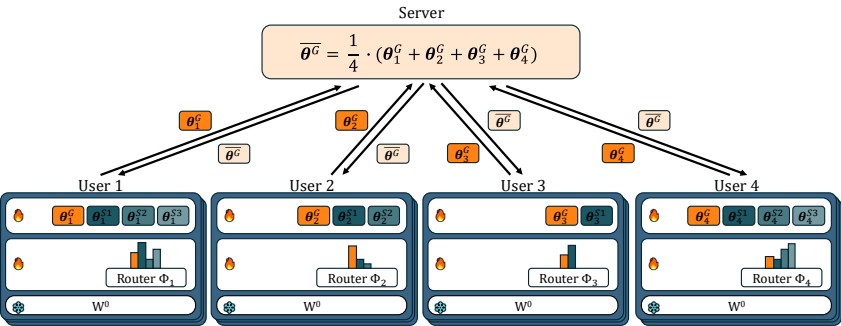

Figure 1: Diagram of our proposed method `CoMiGS` illustrated with a 4-client setup, with a dark blue frame denoting a block. Within each block, generalist experts ($\{\theta_1^G, \theta_2^G, \theta_3^G, \theta_4^G\}$) are aggregated across users, and specialist experts ($\{\theta_1^{Si}, \theta_2^{Si}, \theta_3^{Si}, \theta_4^{Si}\}$) are kept local. Router and expert parameters are updated iteratively using local validation and training datasets.

(2024) explored three personalized collaborator selection protocols, allowing each end user to choose their collaborators. Although these protocols effectively address data heterogeneity, they depend on model aggregation, which can only occur when users share the same model architecture.

There has not yet been a solution to deal with both system heterogeneity and data heterogeneity. Towards this goal, we propose a novel **Co**llaborative learning approach via a **Mi**xture of **G**eneralists and **S**pecialists (CoMiGS). Our approach allows users to share part of the knowledge while keeping some knowledge user-specific, thus providing personalized solutions. We name the shared part *generalists* and the user-specific part *specialists*. Like all previous works, the generalists and specialists are simply LoRA modules. At the same time, as long as the shared part can be aggregated, the user-specific part can be of different sizes, which can be adapted to various device capacities, as illustrated by different numbers of specialists across users in Figure 1.

We integrate the expertise of generalists and specialists using a learned router that determines aggregation weights, following the Mixture-of-Experts (MoE) architecture (Fedus et al., 2022b). As in typical MoE designs for language modeling (Jiang et al., 2024; Fan et al., 2024), we also use tokens as the routing unit. Although users may have different topic preferences or linguistic styles, they still share common tokens in their vocabularies. Our goal is to route these shared tokens to the generalists so they can be jointly learned across users.

The closest work to ours is pFedMoE from Yi et al. (2024) from the vision domain, where each user has a shared homogeneous small feature extractor, a localized heterogeneous feature extractor, and a localized routing network, with routing unit being a semantic unit – an image. The three components are simultaneously updated. Compared to pFedMoE, our method CoMiGS introduces three key updates: 1) we reformulate the learning objective into a bi-level optimization framework, following the inherent hierarchical order of router and expert learning; 2) we refine the routing mechanism by using the smallest unit like a pixel in an image, which is a token; 3) unlike pFedMoE's fixed two-expert limit per user, we support varying numbers of expert modules across users.

In summary, our contributions are as follows:

- We propose a novel approach (CoMiGS) for on-device personalized collaborative fine-tuning of LLMs, introducing an innovative bi-level formulation of the Mixture-of-Experts learning objective. Our approach can effectively tackle distribution shifts in local data.
- Our collaborative framework effectively addresses both *system heterogeneity*, with respect to varying local model architectures, and *data heterogeneity*, concerning diverse local data distributions across users, making it the first model to accomplish both.
- Our framework separates resource heterogeneity from data quantity. Users with larger local datasets benefit from a bigger model, while users with more powerful models but smaller datasets are less prone to overfitting.
- We release a codebase[1] for collaborative LLMs that allows users to easily define their own collaboration strategies, facilitating and advancing future research efforts in this field.

---

[1] Our code base is available at `https://github.com/2025-CoMiGS/codebase`.

## 2 RELATED WORK

**Collaborative Learning for LLMs.** Recently, researchers have been investigating the application of Federated Learning in language tasks. Due to the substantial number of model parameters in LLMs, the research has largely targeted the stages following pre-training, often utilizing parameter-efficient techniques such as adapters. Mohtashami et al. (2023) explored a teacher-student social learning framework to aggregate private-sensitive instructions. Zhang et al. (2023) directly applied FedAvg (McMahan et al., 2017) to aggregate LoRA parameters during instruction tuning, and reported increased performance in downstream tasks. Following that, there are various works focusing on addressing system heterogeneity where users are equipped with different LoRA ranks. HetLoRA (Cho et al., 2023) and FlexLoRA (Bai et al., 2024) provide different ways of aggregating and distributing LoRA modules of heterogenous ranks. However, these approaches are not designed to cope with heterogeneous data on device. In contrast, Sun et al. (2024) found better performances with respect to heterogeneous data can be achieved through freezing LoRA A matrices at initialization; Wagner et al. (2024) proposed personalized solutions that can sufficiently tackle data heterogeneity, through three different collaborator selection mechanisms. Yet for both works, the users must be equipped with the same model architecture. Unlike previous works, our framework deals with both model heterogeneity and data heterogeneity. Moreover, our method offers personalized solutions at a token level, as opposed to the client-level approach in Wagner et al. (2024).

**Mixture of Generalist and Specialist Experts.** Gaspar & Seddon (2022) introduced a fusion of global and local experts for activity prediction based on molecular structures. Each local expert is tailored to a specific chemical series of interest using loss masking, while a global expert is trained across all series. Simultaneously, a routing network learns to assign soft merging scores. This approach yielded superior empirical results compared to single experts. Dai et al. (2024) developed DeepSeekMoE by deterministically assigning every token to "shared" experts, whereas "routed" experts are assigned tokens based on a learnable router. DeepSeekMoE is able to approach the upper bound performance for MoE models. For both works, the notion of shared/global is with respect to input samples, i.e. a shared/global expert should see all input samples. In a collaborative setup, Yi et al. (2024) proposed pFedMoE, where each user has a shared homogeneous small feature extractor, a localized heterogeneous feature extractor, and a localized routing network, with routing unit being an image. The three components are jointly updated in an end-to-end fashion, demonstrating strong performance in the vision domain. Our work builds on the foundations of pFedMoE and extends it to the language domain. Furthermore, we introduce key innovations that enable more effective handling of distribution shifts and achieve a more refined balance between personalization and collaboration.

## 3 METHOD

We aim to improve personalized performance for each user on their target distributions, where distribution shifts can be allowed. Building on the hierarchical insights of MoE learning, we formulate our learning objective into a bi-level optimization problem, where expert parameters are learned using the relatively large-sized training sets, while routing parameters are updated using the small-sized validation sets. We further let experts diversify into generalists and specialists via parameter aggregation or localization, to leverage both collective power and specialized knowledge. As the problem solver, we provide a multi-round gradient-based algorithm.

### 3.1 NOTIONS AND PROBLEM SETUP

Each user has a training set $X_i^{\text{train}}$, a small validation set $X_i^{\text{valid}}$ and a test set $X_i^{\text{test}}$, and the task is next token prediction. The validation set $X_i^{\text{valid}}$ and the test set $X_i^{\text{test}}$ are sampled from the same distribution $\mathcal{P}_i^{\text{target}}$ (note this is a fuzzy concept in the language domain, by the same distribution we mean from the same topic/category). The training set, $X_i^{\text{train}}$, can be sampled from a different distribution than $\mathcal{P}_i^{\text{target}}$. This is to address scenarios where distribution shifts may occur over time, such as changes in topics reflected in the typing data of mobile phone users.

As illustrated in Figure 1, there are two sets of model parameters within each user: expert parameters, denoted as $\boldsymbol{\Theta} = \theta^G \cup \{\theta_i^S\}$, where $\theta^G$ is shared across the users and $\{\theta_i^S\}$ are user-specific specialist parameters; and routing parameters, denoted as $\boldsymbol{\Phi} = \{\phi_i\}$. $i \in \{1, 2, .., N\}$ is the user index. Our ultimate goal is to optimize the average target performance across all users.

Our experts are simply LoRA modules, which approximate model updates $\Delta \boldsymbol{W} \in \mathbb{R}^{m \times n}$ with a multiplication of two low-rank matrices $\boldsymbol{A} \in \mathbb{R}^{m \times r}$ and $\boldsymbol{B} \in \mathbb{R}^{r \times n}$ with rank $r \ll m, n$. $\theta^G$ and $\theta^S$ are disjoint sets of LoRA A and B matrices.

## 3.2 A BI-LEVEL FORMULATION

Essentially, we adopt an MoE architecture, apart from aggregating certain expert parameters. Instead of learning routing and expert parameters simultaneously like the conventional way in LLMs (Zoph et al., 2022; Fedus et al., 2022a), we update the two sets of parameters in an alternating fashion. We observe *a natural hierarchy between the experts and the router*: the assignment of tokens to experts depends on the router's outputs, while the experts' parameters are updated based on the assigned tokens. In this way, the experts' development follows the router's decisions, establishing an inherent leader-follower structure. Following Von Stackelberg (2010), we formulate the hierarchical problem as a bi-level optimization objective in (1). Notably, one of the earliest MoE works (Jordan & Jacobs, 1994), also demonstrates a hierarchical structure, though for a probabilistic interpretation. In contrast, we approach the hierarchical structure from an optimization perspective, formulating the learning process as two nested optimization problems.

$$
\begin{aligned}
\min_{\boldsymbol{\Phi}} \quad & \sum_i \mathcal{L}(f(\boldsymbol{X}_i^{\text{valid}}; \boldsymbol{\Theta}^\star(\boldsymbol{\Phi}), \phi_i), \boldsymbol{X}_i^{\text{valid}}) \\
s.t. \quad & \boldsymbol{\Theta}^\star(\boldsymbol{\Phi}) \in \arg\min_{\boldsymbol{\Theta}} \sum_i \mathcal{L}(f(\boldsymbol{X}_i^{\text{train}}; \theta^G, \theta_i^S, \phi_i), \boldsymbol{X}_i^{\text{train}})
\end{aligned}
\tag{1}
$$

where $\mathcal{L}$ is the language modeling loss. Note we write $\boldsymbol{X}_i$ as the label here, as this is a self-supervised task. Labels are simply shifted inputs. The routing parameters $\boldsymbol{\Phi} = \{\phi_i\}$ are updated based on the validation loss, which reflects the target distribution (outer optimization), while the expert parameters $\boldsymbol{\Theta} = \theta^G \cup \{\theta_i^S\}$ are updated using the training loss (inner optimization). This formulation further brings in the following benefits: 1) routing parameters are smaller in size, making them easier to overfit. By separating the two losses, the routing parameters can be updated less frequently using the smaller validation set (a visual evidence of less frequent router update leading to improved performance is provided in Figure 7 in the Appendix); 2) this approach handles situations where target distributions differ from training distributions more effectively, as the router outputs (i.e., how the experts should be weighted) can be tailored to specific tasks.

## 3.3 OUR ALGORITHM

To solve (1), we use a multi-round gradient-based algorithm as shown in Alg.1, where only generalist parameters are shared and aggregated, and specialist and router parameters are updated locally. While the scheme requires a server, it can alternatively be implemented in a serverless all2all fashion, which requires $N$ times more communication overhead and we do not further pursue this here.

**Alternating Update of $\boldsymbol{\Theta}$ and $\boldsymbol{\Phi}$:** Alternating update of two sets of parameters is a standard way to solve bi-level optimization problems (Chen et al., 2021). In between two communication rounds, we perform alternating updates of expert and routing parameters using local training and validation sets separately. The updates optimize the objectives given in (2) and (3) respectively. Since the updates of $\boldsymbol{\Theta}$ and $\boldsymbol{\Phi}$ are disentangled, they do not need to be updated at the same frequency. The routing parameters are smaller in size and thus can be updated less frequently. When updating model parameters, we include an additional load-balancing term as in Fedus et al. (2022a), which is standard in MoE implementation and encourages even distribution of token assignments to experts. A discussion over the load balancing term is included in Appendix C.4. It is observed that a load-balancing term can improve test performance compared to not having one. However, directing more tokens to the generalists has no noticeable effect.

**Update of $\theta^G$ and $\theta_i^S$:** The update of generalist parameters $\theta^G$ follows a standard FedAvg scheme, through aggregating model parameters. Specifically, we simultaneously update both $\theta^G$ and $\theta_i^S$ by optimizing equation (2), which results in $\theta_i^G$ and $\theta_i^S$. A parameter aggregation is then performed on the user-specific $\theta_i^G$ via a trusted server to establish a shared $\theta_G$ across all users. In the next round, each user replaces their $\theta_i^G$ with the global $\theta_G$, while their $\theta_i^S$ remains locally updated.

---

**Algorithm 1** Pseudo code of our proposed algorithm

---

**Input:** Expert parameters $\{\theta_{i,0}^G, \theta_{i,0}^S\}$, routing parameters $\{\phi_{i,0}\}$. Local training data and validation data $\{\boldsymbol{X}_i^{\text{train}}, \boldsymbol{X}_i^{\text{valid}}\}, i \in \{1, 2, .., N\}$. Communication round $T$ and routing update period $\tau$. Load balancing weight $\lambda$.

**for** $t = 1, ..., T$ **do**
    Server aggregates generalist parameters: $\theta_{t-1}^G = \frac{1}{N} \sum_i \theta_{i,t-1}^G$
    **for** $i \in [0, N)$ **do**
        Users download aggregated generalist weights and
        prepare model parameters for training $\{\theta_{t-1}^G, \theta_{i,t-1}^S, \phi_{i,t-1}\}$
        Do gradient steps on $(\theta_{t-1}^G, \theta_{i,t-1}^S)$ towards minimizing (2) and get $(\theta_{i,t}^G, \theta_{i,t}^S)$

$$\min_{\theta_i^G, \theta_i^S} \mathcal{L}(f(\boldsymbol{X}_i^{\text{train}}; \theta_i^G, \theta_i^S, \phi_{i,t-1}), \boldsymbol{X}_i^{\text{train}}) + \lambda \cdot \mathcal{L}_i^{\text{LB}}(\boldsymbol{X}_i^{\text{train}}; \theta_i^G, \theta_i^S, \phi_{i,t-1}) \quad (2)$$

        **if** $t\%\tau = 0$ **then**
            Do gradient steps on $\phi_{i,t-1}$ towards minimizing (3) and get $\phi_{i,t}$

$$\min_{\phi_i} \mathcal{L}(f(\boldsymbol{X}_i^{\text{valid}}; \theta_{i,t}^G, \theta_{i,t}^S, \phi_i), \boldsymbol{X}_i^{\text{valid}}) + \lambda \cdot \mathcal{L}_i^{\text{LB}}(\boldsymbol{X}_i^{\text{valid}}; \theta_{i,t}^G, \theta_{i,t}^S, \phi_i) \quad (3)$$

        **end if**
    **end for**
    Each device $i \in \{1, 2, .., N\}$ sends generalist weights $\theta_{i,t}^G$ to the server
**end for**
**Return:** Expert parameters $\{\theta_{i,T}^G, \theta_{i,T}^S\}$ and routing parameters $\{\phi_{i,T}\}$

---

**Theorem 3.1 (Convergence Result)** *If 1) the two loss functions on training and validation sets share the same global minimum $(\boldsymbol{\Theta}^\star, \phi_i^\star)$ for all $i \in [N]$, 2) for any $\boldsymbol{\Theta}', \phi_i', \mathcal{L}(f(\boldsymbol{X}_i^{valid}; \boldsymbol{\Theta}', \phi_i), \boldsymbol{X}_i^{valid})$ and $\mathcal{L}(f(\boldsymbol{X}_i^{train}; \boldsymbol{\Theta}, \phi_i'), \boldsymbol{X}_i^{train})$ are strongly convex in $\phi_i$ and $\boldsymbol{\Theta}$ respectively, following the alternating update in Algorithm 1, we have $(\boldsymbol{\Theta}_t, \phi_{i,t})$ converge to $(\boldsymbol{\Theta}^\star, \phi_i^\star)$ linearly.*

The proof is presented in Appendix G.

## 4 EXPERIMENTS

### 4.1 SETUP

#### 4.1.1 DATASETS

We selected three diverse distributed datasets to demonstrate the efficacy of our proposed algorithm:

    i) *Multilingual Wikipedia*: This dataset constitutes Wikipedia articles in four languages: German, Dutch, French, and Italian. We take German, French and Italian from Wikimedia-Foundation, and Dutch from Guo et al. (2020);

    ii) *SlimPajama*: We pick the following four categories – StackExchange, Github Codes, ArXiv, Book from Soboleva et al. (2023);

    iii) *AG News*: This dataset covers News from categories of World, Sports, Business, and Sci/Tech (Zhang et al., 2016).

Opting for the most challenging scenario, each user is assigned a unique category, as shown in Figure 8, where users can have varying data quantities. Given our emphasis on next token prediction, we anticipate shared predictions among users while maintaining category-specific distinctions. For details of our user data splits, please refer to Appendix B. We further create the following two scenarios to showcase the wide applicability of our method:

**In-Distribution Tasks.** For each user, we construct validation and test datasets that follow the same distribution as the training data. We address two scenarios in this context: (i) variation in language usage across users (Multilingual Wikipedia), and (ii) variation in topic coverage across users (SlimPajama).

**Out-of-Distribution Tasks.** For each user, we create validation and test datasets from a distribution different from the training data. During training, each user is assigned a single News category from AG News, but their validation and test sets consist of a uniform mixture of all categories. This approach accounts for potential shifts in topics within users.

### 4.1.2 EXPERIMENTAL DETAILS

Our base models are the GPT2 model with 124M parameters and Llama 3.2 model with 1B parameters, which are suitable for on-device deployment[2]. We incorporate LoRA modules into every linear layer, including MLP and Self-Attention Layers, following the recommendations of Fomenko et al. (2024), specifically in the [attn.c_attn, attn.c_proj, mlp.c_fc, mlp.c_proj] layers. A routing mechanism is exclusively implemented atop MLP layers. This means that each attention layer has only one LoRA expert applied to it, which is always aggregated during synchronization. The number of LoRA experts in MLP blocks depends on the local resource abundance. For more experimental details, we refer readers to Appendix B.

### 4.2 DATA-DRIVEN SELECTION: GENERALIST VS. SPECIALIST

We start by equipping users with the same model architecture locally, to illustrate the effectiveness of our hierarchical learning of routing and expert parameters. We compare our one generalist one specialist (CoMiGS-1G1S) method to the following baselines. In order to match the trainable parameter count of our method, we use 2 times LoRA modules within each user.

   i) Upper and lower bounds:
   - Pretrained: A pretrained GPT-2 model using weights from OpenAI.
   - Centralized: A single model trained using data from all users. (Note this method is an unrealistic baseline as data cannot leave the devices due to privacy concerns.)

  ii) Baselines:
   - Local: Training individually using only local data without collaboration.
   - FedAvg: Aggregating LoRA parameters across users using uniform weights, which is equivalent to applying FedAvg (McMahan et al., 2017).
   - PCL: Aggregating LoRA parameters using a client-level collaboration graph. The graph is updated using validation performances. (Strategy 2 in Wagner et al. (2024)).
   - pFedMoE: We directly apply the method from Yi et al. (2024) in the language domain where we update routing and expert parameters at the same time and choose tokens as a routing unit.

 iii) Ablations:
   - CoMiGS-2S: Both of the LoRA experts are specialists, meaning their weights are neither shared nor aggregated. The routing parameters are updated using a separate validation set like in CoMiGS-1G1S.
   - CoMiGS-2G: Both of the LoRA experts are generalists, meaning their weights are always shared and aggregated. The routing parameters are updated using a separate validation set like in CoMiGS-1G1S.

### 4.2.1 RESULT ANALYSIS

The comparison between our method and the baseline methods is summarized in Table 1.

**Effectiveness of our routing mechanism:** Depending on the dataset, either CoMiGS-2G or CoMiGS-2S achieves the highest performance. The key distinction compared to Local or FedAvg is the existence of a layer-wise router, which weighs the two generalists or two specialists for each token according to the validation performances, as opposed to assigning equal weights. This emphasizes that even with the same expert knowledge, the way it's combined is crucial. Moreover, pFedMoE, despite having a learned router as well, underperforms our method, even in the in-distribution scenario. The reason is that the routing parameters are updated simultaneously with the expert parameters using the training set, and thus cannot effectively adapt to the target distribution.

---

[2]We adopt the codes from https://github.com/karpathy/nanoGPT and https://github.com/danielgrittner/nanoGPT-LoRA, https://github.com/pjlab-sys4nlp/llama-moe

| | In Distribution | | Out of Distribution |
|---|---|---|---|
| | *Multilingual* | *SlimPajama* | *AG News* |
| *Pretrained* | 156.12 | 37.19 | 90.65 |
| *Centralized* | 55.41 (0.12) | 19.53 (0.14) | 28.19 (0.52) |
| *Local* | 54.38 (0.32) | 26.95 (0.14) | 41.46 (0.06) |
| *FedAvg* | 58.80 (0.34) | 23.27 (0.05) | 31.84 (0.02) |
| *PCL* | 54.53 (0.19) | 26.99 (0.19) | 32.25 (0.12) |
| *pFedMoE* | 52.27 (0.17) | 25.40 (0.09) | 38.72 (0.21) |
| *CoMiGS - 2S (ours)* | 46.36 (0.16) | 22.51 (0.08) | 35.81 (0.13) |
| *CoMiGS - 2G (ours)* | 58.31 (0.17) | 21.36 (0.01) | 31.18 (0.05) |
| *CoMiGS - 1G1S (ours)* | 47.19 (0.10) | 21.79 (0.04) | 33.53 (0.03) |

Table 1: Mean test perplexity over the users with homogeneous models, averaged across 3 seeds. Mean (std) with a rank locator for the mean (the lower the better). **Green** denotes the best performing methods and **red** denotes our method. An extended version of the table can be found in Table 4. A replicated experiment using Llama3.2 (1B) base model can be found in Table 6.

**Token-level collaborative decisions outperform Client-level:** Compared to the state-of-the-art baseline PCL, as proposed by Wagner et al. (2024), our method demonstrates a clear performance improvement. PCL assigns a pairwise collaboration weight between users by evaluating how well user $i$'s model performs on user $j$'s validation set. On the two in-distribution tasks, PCL exhibits performance similar to Local, where the learned collaboration matrices are nearly identity matrices, thereby limiting effective collaboration between users. Our method, in contrast, decides the collaboration pattern based on each input token, and thus can harness the collective power more effectively.

**The necessity of the co-existence of generalists and specialists:** The performances of CoMiGS-2G and CoMiGS-2S are not consistent across the different scenarios, while our CoMiGS-1G1S can always closely track the best-performing model, which is clearly shown in Table 1 and visualized in Figure 9. Depending on the task type, generalists and specialists alone may not be sufficient. A balanced combination of personalization and collaboration is required, and our approach achieves this balance effectively.

**Computational and communication overhead:** Please refer to Appendix B.1.

### 4.2.2 ROUTING ANALYSIS

**Token-wise analysis:** We further present a token-level routing result visualization on models fine-tuned with SlimPajama dataset in Figure 2: The first two users are fine-tuned with very specific math and programming texts, and they tend to utilize the generalist more in the last layer. Function words ("and", "a", "on" "the" etc) are more routed to generalists, as expected. This can be seen in the top right panel of Figure 2. It is important to note that only the top choice is highlighted here. The abundance of blue does not imply that generalist experts play no role in predicting the next token. To see this, compared to when only specialists are present (CoMiGS-2S), our CoMiGS-1G1S gives more consistent results. More detailed token-wise routing result visualization including out-of-distribution tasks can be seen in Appendix F. When dealing with out-of-distribution texts, there is an increasing tendency to seek for generalists, as shown in the off-diagonal entries in Figure 14-19.

**Layer-wise analysis:** Figure 3 depicts the evolution of averaged layer-wise router outputs for the generalist and specialist experts on the *out-of-distribution* task, comparing CoMiGS-1G1S and pFedMoE. As training progresses, CoMiGS-1G1S undergoes a *phase transition*: the layer-wise routers initially favor generalists but gradually shift towards specialists. This shift is not observed in pFedMoE, highlighting the critical role of our routing mechanism in handling out-of-distribution tasks. Additionally, we notice different layers converge to a different expert score distribution. When applying our CoMiGS-1G1S, for each user, there are always certain layers where the routers consistently prefer generalists, which aligns with the fact that our target distribution is a union of all local training distributions. This phenomenon no longer occurs with in-distribution tasks, as shown in Figure 10.

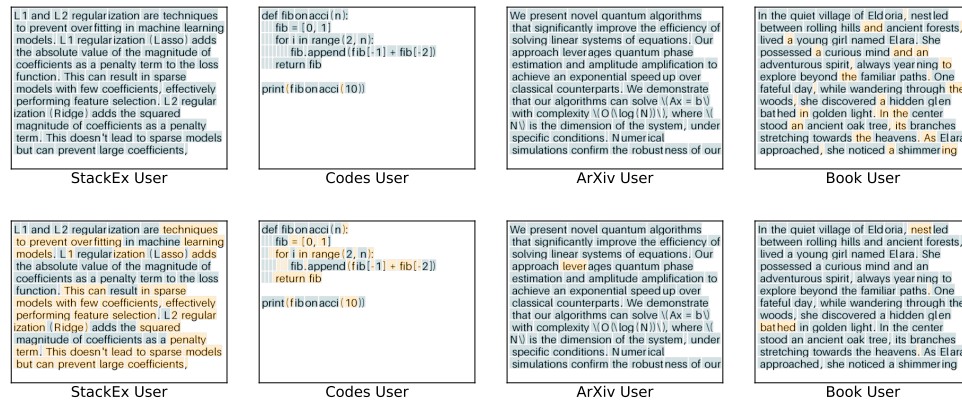

Figure 2: Visualization of in-distribution token-level routing results for `CoMiGS-1G1S` trained on SlimPajama. Tokens are colored with the Top1 expert choice at the first layer (top) and last layer (bottom). Orange denotes the generalist and blue denotes the specialist. Texts are generated by ChatGPT. Further colored text plots are provided in Appendix F.

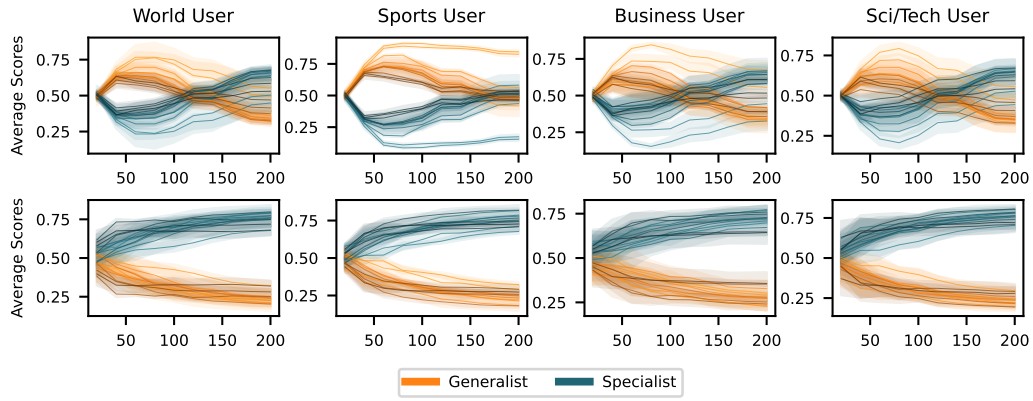

Figure 3: Expert Scores for the *generalist* expert and the *specialist* expert, averaged across all tokens and multiple batches for the out-of-distribution task (AG News), with x-axis being the number of iterations. Upper row: our `CoMiGS-1G1S`, bottom row: `pFedMoE`. Darker colors represent deeper layers. Expert score plots for in-distribution tasks can be seen in Figure 10 .

### 4.3 ADAPTATION TO RESOURCE HETEROGENEITY

#### 4.3.1 BASELINE COMPARISON

In this section, our focus is to deal with system heterogeneity, where users can have different numbers of experts. We still keep one generalist expert, but the number of specialists can vary across the users (our method is denoted as One-Generalist-X-Specialists, named `CoMiGS-1GXS`). It's important to note that the richness of computational resources doesn't always correlate with the complexity of local data. For instance, some users may have ample computational resources but local data in small quantities. In such cases, a crucial objective is to prevent overfitting due to redundant model-fitting abilities.

We compare our approach to two state-of-the-art baselines: `HetLoRA` from Cho et al. (2023) and `FlexLoRA` from Bai et al. (2024), both of which adapt LoRA ranks based on the resource capacity of each user. `HetLoRA` aggregates LoRA matrices $A$ and $B$ by zero-padding to the maximum rank and then distributes them back using rank truncation. In contrast, `FlexLoRA` first reconstructs model updates $\Delta W$ and redistributes the aggregated updates using SVD. We compare our method to these baselines by matching the number of fine-tunable parameters, measured as both active and full parameters. For example, to match the full parameter count of `CoMiGS-1GXS` with $(4, 2, 2, 2)$

LoRA experts (rank 8), LoRA modules of ranks $(32, 16, 16, 16)$ would be required. With Top2 routing, to match the active parameter count, each user would need LoRA modules of rank 16.

Our results, presented in Table 2, are based on allocating different computational resources to users, with resource availability decoupled from local task complexity. We find that our method outperforms the baseline methods most of the time, regardless of whether we match the full parameter count or the active parameter count. This advantage stems from the fact that both `HetLoRA` and `FlexLoRA` average model parameters across users without allocating parameters for local adaptations, focusing on building a strong generalist model. In contrast, our approach adaptively integrates both generalist and specialist knowledge, excelling in scenarios where specialized knowledge is crucial.

|  |  | **Ours** | **HetLoRA** | | **FlexLoRA** | |
|---|---|---|---|---|---|---|
|  |  | CoMiGS-1GXS | Active | Full | Active | Full |
| *In Distribution* | **Multilingual** |  |  |  |  |  |
|  | *(2,2,4,4)* | 46.48 (0.16) | 57.76 (0.10) | 58.60 (0.20) | 77.65 (0.20) | 77.85 (0.26) |
|  | *(4,4,2,2)* | 47.24 (0.09) | 57.76 (0.10) | 59.14 (0.04) | 77.65 (0.20) | 76.29 (0.17) |
|  | **SlimPajama** |  |  |  |  |  |
|  | *(2,4,4,2)* | 22.10 (0.17) | 23.33 (0.10) | 23.15 (0.09) | 22.97 (0.11) | 22.99 (0.08) |
|  | *(4,2,2,4)* | 22.28 (0.09) | 23.33 (0.10) | 23.17 (0.09) | 22.97 (0.11) | 22.99 (0.09) |
| *Out of Distribution* | **AG News** |  |  |  |  |  |
|  | *(4,2,2,2)* | 33.66 (0.07) | 31.58 (0.14) | 31.95 (0.13) | 36.45 (0.06) | 36.49 (0.17) |
|  | *(2,4,4,4)* | 34.22 (0.09) | 31.58 (0.14) | 32.52 (0.19) | 36.45 (0.06) | 36.40 (0.08) |

Table 2: Mean test perplexity (std) over users with heterogeneous models, averaged across 3 seeds, with red being the top1 method. For example, $(4, 2, 2, 2)$ means in our `CoMiGS-1GXS` setup users have 4, 2, 2, and 2 experts, respectively, and in the two baselines, all users have rank 16 to match active parameter count, or ranks 32, 16, 16, and 16 to match full parameter count.

### 4.3.2 ANALYSIS RELATED TO LOCAL DATA QUANTITIES

In this section, we further separate resource abundance from data quantity. It is observed that our approach is more robust to overfitting due to the regularizing effect of the generalist, while at the same time better fitting local data through the incorporation of specialist knowledge.

We conduct experiments using Multilingual Wikipedia dataset, where we allocate low data quantities to German and Dutch users, and high data quantities to French and Italian users, as shown in Figure 8. In practice, users may not know their local data complexity, leading to a potential mismatch in resource allocation relative to data quantity. To simulate such scenarios, we allocate model capabilities—measured by the number of LoRA modules per user—either positively or negatively correlated with their local data size. It is important to note that one generalist is always assigned, and resource abundance is only reflected in the number of specialists.

**More Specialists Help with Higher Data Quantity.** French and Italian users consistently benefit from having more specialists locally, as their test perplexities decrease when the number of specialists increases from 1 to 3 to 7. This suggests that when sufficient local training data is available, adding more specialists leads to improved performance.

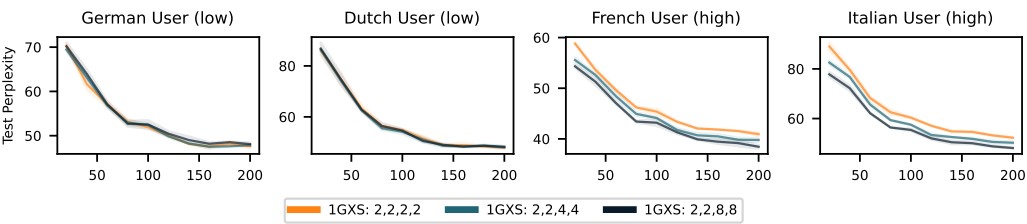

Figure 4: Test Perplexity vs. the number of iterations. Low and high denote the relative data quantity among users. The numbers in the legend indicate the number of experts $n_i$ within each user. Top-2 routing is performed.

**Generalists Help to Prevent Redundant Specialists from Over-Fitting.** For users with low data quantities, local model training with just two LoRA modules already results in overfitting (a trend

observed in Figure 9). Our goal here is to prevent overfitting. Figure 5 demonstrates that our method succeeds to surpress overfitting, even when fine-tuning twice or four times as many expert parameters. We attribute this to the existence of the generalists.

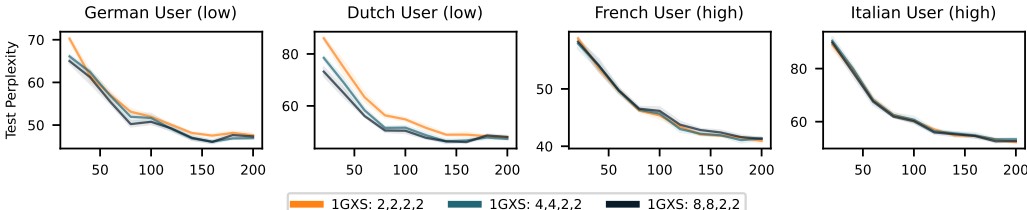

Figure 5: Test Perplexity vs. the number of iterations. Low and high denote the relative data quantity among users. The numbers in the legend indicate the number of experts $n_i$ within each user. Top-2 routing is performed. German and Dutch Users despite having high resources locally, do not overfit on their small-sized local data.

**Specialists Can Benefit Generalists.**    What happens if users can only support a maximum of one expert? In our setup, such users must rely on the generalist expert when participating in collaboration. Interestingly, even when their collaborators are allocated more specialists, low-resourced users with only one generalist still benefit from the refined role diversification between generalists and specialists. As a result, the generalists become more powerful, as demonstrated in Figure 6.

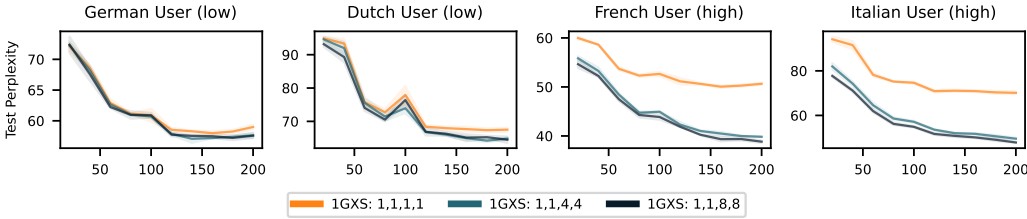

Figure 6: Test Perplexity vs. the number of iterations. German and Dutch Users, despite having only one expert locally, still benefit from their collaborators having more experts, thereby enhancing the generalist's performance. The numbers in the legend indicate the number of experts, $n_i$, within each user. Top-2 routing is applied when $n_i \geq 2$.

We provide an additional example of the impact of local data quantities in Appendix E using SlimPajama dataset. Similar conclusions can be drawn from our empirical results. However, there is a limit to how much generalists can help prevent overfitting when the local tasks are easy.

## 5    CONCLUSIONS AND FUTURE DIRECTIONS

We propose a novel framework for on-device personalized collaborative fine-tuning of LLMs, grounded in an innovative bi-level formulation of the Mixture-of-Experts learning objective. Our fine-grained integration of generalist and specialist expert knowledge achieves superior performance in balancing personalization and collaboration within Federated LLMs.

Furthermore, our framework is the first to address both system and data heterogeneity in collaborative LLM training. It also decouples local data quantity from resource availability, allowing high-resourced users to leverage larger datasets for improved performance while remaining resilient against overfitting in low-data scenarios.

An interesting future direction to explore is adopting our framework for collaborative instruction tuning of larger LLMs and evaluating its performance on downstream tasks. While our paper focused on a single generalist, it is possible to include multiple generalists, and their impact on performance remains to be seen. We hope our work paves the way for a new direction in on-device collaborative LLMs.

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

## A    LIMITATIONS AND SOCIETAL IMPACT

**Limitations.**    Compared to FedAvg type of methods, our method requires extra gradient steps on routing parameters and memory storage for the routing parameters. Since a routing network is usually a one-layer MLP, the extra cost in computation and storage is relatively small.

The robust performance of our method relies on the fact that we update routing parameters and expert parameters on two independent losses. This means we need some validation samples independent from training samples. When local data size is minimal, this can be an issue.

Our method, while generally robust, still has a tendency towards overfitting when there is a significant mismatch between local resource abundance and data complexity, similar to other methods.

**Societal Impact.**    We offer a collaboration framework for edge devices, aiming to enable smaller devices to leverage large language models (LLMs) despite limited resources and data availability. Our approach enhances fairness and mitigates privacy concerns by ensuring data remains on end devices. The privacy aspects can further be enhanced by differential private aggregation of generalist weights, which we do not pursue here.

The robustness towards attackers is beyond the scope of our work. Our collaboration framework has no guarantee of resilience towards Byzantine attackers, which could potentially lead to misuse by certain parties.

## B    EXTRA EXPERIMENTAL DETAILS

### B.1    COMPUTATIONAL AND COMMUNICATION OVERHEAD

**Computational overhead**: During a forward pass, on top of the base pre-trained GPT2 model ( 32 GFlops), fine-tuning using FedAvg with two sets of LoRA modules adds extra 490 MFLOPs (+ 1.53%), while the typical FedAvg with one set of LoRA models adds extra 166 MFlops (+ 0.52%). Our CoMiGS-1G1S adds 495 extra MFLOPs (+ 1.55%, 490MFLOPS from the experts and 5MFLOPs from the router). The FLOPs are approximated following Appendix B of Chowdhery et al. (2023). The extra computational complexity is almost neglectable in comparison to the base model.

**Extra memory requirement**: Compared to storing the LoRA matrices, the extra memory storage from the router is 0.035 MB, assuming bfloat16 training.

**Communication costs**: Since specialists and routers stay locally within each device, the only weight to communicate is from the generalist experts. As we conduct fine-tuning with bfloat16, in each communication round, each device only needs to communicate 1.41 MB of generalist weights, which we do not consider a big value.

### B.2    TRAINING DETAILS

Following Kalajdzievski (2023), we choose $\gamma$ to be a rank-stabilized value, a technique which helps stabilize gradient norms. $\alpha$ and the rank $r$ are hyper-parameters to choose from. The LoRA modules function as follows:

$$\boldsymbol{W} = \boldsymbol{W}^0 + \gamma \cdot \boldsymbol{A}\boldsymbol{B}, \qquad \gamma = \frac{\alpha}{\sqrt{r}} \tag{4}$$

All our experiments except the centralized ones were conducted on a single A100-SXM4-40GB GPU. The centralized learning baseline experiments were conducted on a single A100-SXM4-80GB GPU, as a batch size of 64*4 requires a larger storage capacity.

We use a constant learning rate of $2 \times 10^{-3}$ for updating routing parameters and a $2 \times 10^{-3}$ learning rate with a one-cycle cosine schedule for expert parameters during fine-tuning. The LoRA rank $r$ is set to 8 unless otherwise specified, with LoRA alpha $\alpha$ set to 16, following the common practice of setting alpha to twice the rank (Raschka, 2023). A load balancing weight $0.01$ is always applied.

For AG News and Multilingual Wikipedia data splits, we conduct 20 communication rounds. For SlimPajama data splits, due to greater category diversity, we conduct 50 communication rounds.

Between each pair of communication rounds, there are 10 local iterations. In each iteration, a batch size of 64 is processed with a context length of 128. We set the routing update period to 30 iterations, and every time we update routing parameters, we do 10 gradient steps on the validation loss. The choice of the hyperparamters is from a sweep run and we provide the evidence in Figure 7.

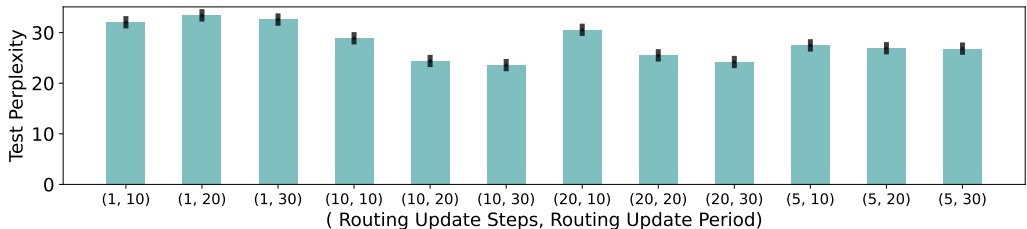

Figure 7: Sweep results on SlimPajama data splits. We ablate the impact of the update period ($\tau$) and the number of update steps ($s$) on model performance.

### B.3 DATA DISTRIBUTION

The dataset distribution and number of tokens within each user are shown in Figure 8 and Table 3 respectively.

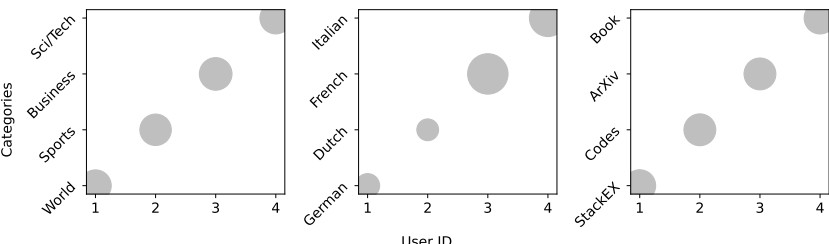

Figure 8: The data splits across users, with bubble size denoting the relative size of the local dataset.

|  |  | User 1 | User 2 | User 3 | User 4 |
|---|---|---|---|---|---|
| **Multilingual** | Training | 557'662 | 407'498 | 556'796 | 451'584 |
|  | Validation | 300'764 | 216'318 | 220'071 | 165'984 |
|  | Test | 229'720 | 219'741 | 210'570 | 172'547 |
| **SlimPajama** | Training | 1'000'000 | 1'000'000 | 1'000'000 | 1'000'000 |
|  | Validation | 200'000 | 200'000 | 200'000 | 200'000 |
|  | Test | 200'000 | 200'000 | 200'000 | 200'000 |
| **AG News** | Training | 761'924 | 756'719 | 814'131 | 771'460 |
|  | Validation | 48'809 | 48'730 | 50'398 | 48'249 |
|  | Test | 48'167 | 47'721 | 48'344 | 49'377 |

Table 3: Number of tokens in each dataset splits

## C MORE TABLES AND FIGURES

### C.1 LEARNING CURVES OF DIFFERENT METHODS

### C.2 EXTENDED BASELINE COMPARISON

An extended version of Table 1 is presented in Table 4. In this extension, we incorporate two additional ablations: 1) Integration of a routing mechanism, updated simultaneously with the expert

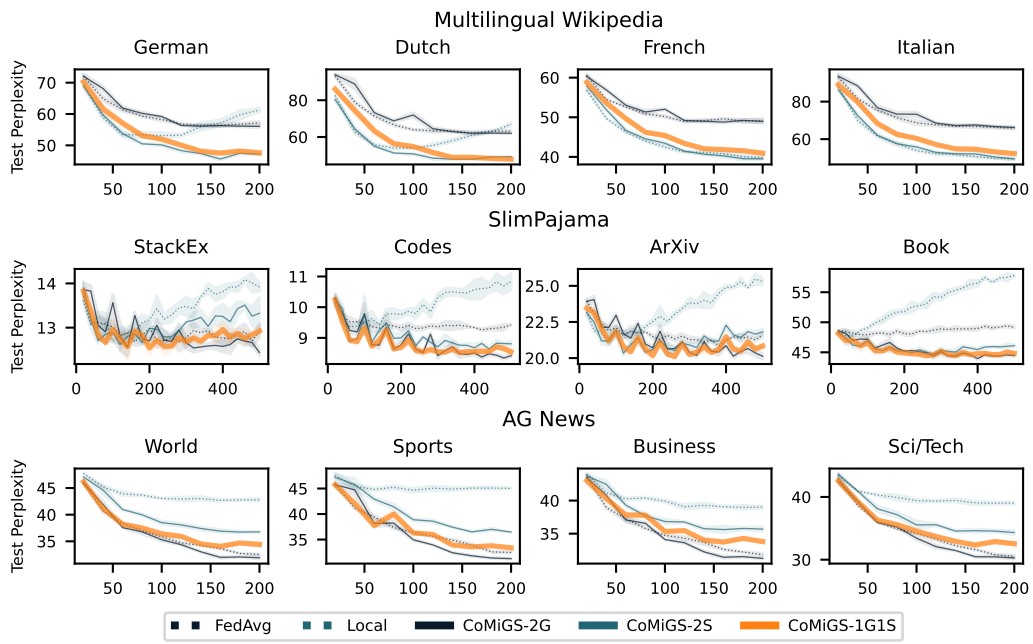

Figure 9: Test Perplexity during training for all the three datasets: our method closely follows the best performing method

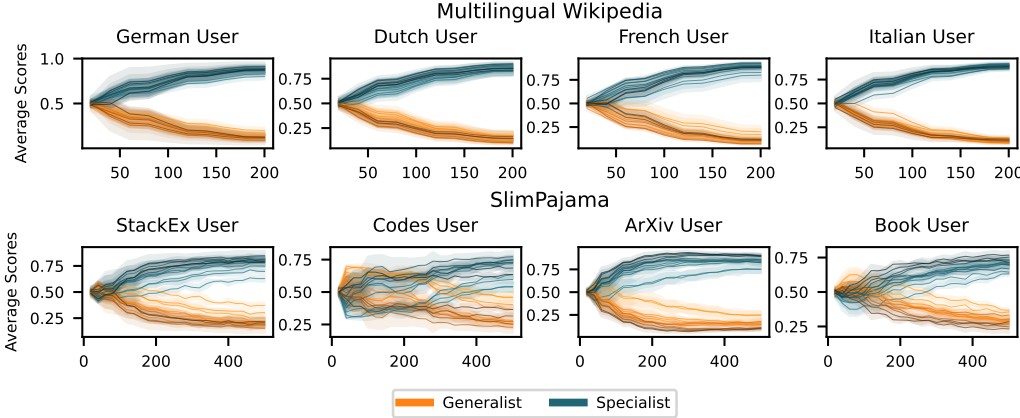

Figure 10: Expert Scores for the *generalist* expert and the *specialist* expert from our `CoMiGS-1G1S` method, averaged across all tokens and multiple batches for the in-distribution task, with x-axis being the number of iterations. Darker colors represent deeper layers.

networks; 2) Iterative updates alternating between routing and expert parameters, with the routing parameters updated using newly-sampled training batches instead of a dedicated validation set. Moreover, we include another baseline method FFA-LoRA from Sun et al. (2024), where the LoRA A matrices are fixed at initialization.

Notably, the comparison between scenarios ii) and iii) reveals minimal disparity, underscoring the significance of having an independent validation set exclusively for routing parameter updates.

## C.3 HETLORA

Analogously to the baseline experiment comparison in FlexLoRA (Bai et al., 2024), we use $\gamma = 0.99$ as pruning strength and sweep the regularization parameter in $\{5 \times 10^{-2}, 5 \times 10^{-3}, 5 \times 10^{-4}\}$.

| | In Distribution | | Out of Distribution |
| --- | --- | --- | --- |
| | *Multilingual* | *SlimPajama* | *AG News* |
| i) Without routing | | | |
| *Pretrained* | 156.12 | 37.19 | 90.65 |
| *Centralized* | 55.41 (0.12) | 19.53 (0.14) | 28.19 (0.52) |
| *Local* | 54.38 (0.32) | 26.95 (0.14) | 41.46 (0.06) |
| *FedAvg* | 58.80 (0.34) | 23.27 (0.05) | 31.84 (0.02) |
| *FFA-LoRA* | 57.83 (0.13) | 23.42 (0.069) | 31.60 (0.14) |
| *PCL* | 54.53 (0.19) | 26.99 (0.19) | 32.25 (0.12) |
| ii) Update routing and expert params simultaneously on training loss | | | |
| *Local-MoE* | 55.27 (0.40) | 27.16 (0.16) | 41.49 (0.01) |
| *FedAvg-MoE* | 56.77 (0.37) | 23.32 (0.07) | 32.24 (0.08) |
| *pFedMoE* | 52.27 (0.17) | 22.91 (0.18) | 38.72 (0.21) |
| iii) Alternating update routing params on newly sampled batches from training set | | | |
| *Local-MoE - tr* | 53.78 (0.33) | 27.78 (0.06) | 41.46 (0.03) |
| *FedAvg-MoE - tr* | 59.39 (0.13) | 23.00 (0.01) | 31.70 (0.16) |
| *CoMiGS - tr* | 50.86 (0.14) | 25.45 (0.01) | 38.93 (0.08) |
| iv) Alternating update routing params on a validation set | | | |
| *CoMiGS - 2S* | 46.36 (0.16) | 22.51 (0.08) | 35.81 (0.13) |
| *CoMiGS - 2G* | 58.31 (0.17) | 21.36 (0.01) | 31.18 (0.05) |
| *CoMiGS - 1G1S* | 47.19 (0.10) | 21.79 (0.04) | 33.53 (0.03) |

Table 4: Mean test perplexity over users with homogenous models, averaged across 3 seeds. Mean (std) with a rank locator for the mean (the lower the better). Green denotes the best performing methods and red denotes our method.

## C.4 IS THE STANDARD LOAD BALANCING LOSS SUFFICIENT?

The standard load balancing loss encourages equal assignment of tokens to each expert. When the number of experts gets larger, there might not be enough tokens routed to the generalists, which might lead to a under-developed general knowledge. We will verify if this is indeed true.

To encourage enough tokens to be routed to the generalist expert such that more general knowledge can be developed, we modify our load-balancing loss by introducing importance weighting. As we separate the 0-th expert to be the generalist expert and conduct Top-2 routing, the modified load balancing loss is as follows:

$$\mathcal{L}_i^{\text{LB}} = \frac{1}{(n_i - 1)^2 + 1} \cdot f_0 \cdot P_0 + \sum_{j=1}^{n_i - 1} \frac{n_i - 1}{(n_i - 1)^2 + 1} \cdot f_j \cdot P_j \tag{5}$$

where

$$f_j = \frac{1}{T} \sum_{x \in \mathcal{B}} \mathbb{1}\{j \in \text{Top2 indices of } p(x)\} \qquad P_j = \frac{1}{T} \sum_{x \in \mathcal{B}} p_j(x) \tag{6}$$

$j$ is the expert index and $p(x) = [p_j(x)]_{j=1}^{n_i}$ is the logit output from the routing network for a specific token $x$. The idea is that one of the top 2 tokens should always be routed to the generalist expert, i.e. the 0-th expert. Thus, $\frac{p_0}{1/2}$ should be equal to $\frac{p_i}{1/2(n_i - 1)}$ for $i \neq 0$. As the original load balancing loss encourages uniform distribution, this modification encourages the generalist expert to have a routing probability of 0.5 on expectation. Note that when $n_i = 2$, this $\mathcal{L}_i^{\text{LB}}$ is the same as the original load balancing loss as proposed in Fedus et al. (2022a).

We present the results in Table 5: in both scenarios, whether users have the same or different numbers of experts, including a load-balancing term leads to a slight improvement compared to omitting it. However, encouraging more tokens to be routed to the generalists does not make a significant difference.

|  | No LB | LB (uniform) | LB (generalist-favored) |
|---|---|---|---|
| AG News (homo) | 33.69 (0.21) | 33.53 (0.03) | 33.53 (0.03) |
| AG News (hetero) | 34.31 (0.05) | 34.28 (0.11) | 34.22 (0.09) |
| Multi-Wiki (homo) | 47.31 (0.15) | 47.19 (0.10) | 47.19 (0.10) |
| Multi-Wiki (hetero) | 46.36 (0.16) | 46.15 (0.04) | 46.48 (0.16) |
| SlimPajama (homo) | 21.77 (0.02) | 21.79 (0.04) | 21.79 (0.04) |
| SlimPajama (hetero) | 22.15 (0.07) | 22.10 (0.11) | 22.10 (0.17) |

Table 5: Test perplexity with different load balancing terms with (hetero) or without (homo) system heterogeneity.

## D   LLAMA3.2 (1B) EXPERIMENTS

We replicate our experiments of Table 1 with a Llama 3.2 (1 B) model in Table 6. Given the extensive pre-training of LLAMA 3 models on over 15 trillion tokens from public sources (Meta, 2024a), and the multilingual capabilities of LLAMA 3.2 (1B) (Meta, 2024b), fine-tuning on multilingual Wikipedia or SlimPajama resulted in negligible improvements likely due to significant overlap with the pre-training data corpus.

Therefore, in the Llama3.2 (1B) experiments we introduce a new fine-tuning dataset, which is derived from Common Corpus (pleias, 2024) - specifically, the YouTube-Commons, Latin-PD, and TEDEUTenders collections - and the Harvard USPTO dataset (Suzgun et al., 2022). Following our previous methodology, each client is assigned one of the datasets to maximize heterogeneity. We use this dataset to model the in-distribution scenario. Additionally, we reduced the number of training iterations for the AG News experiment.

Our results on the Common Corpus-based dataset, which emphasizes domain-specific language and structure, demonstrate that our CoMiGS-1G1S method can outperform local training and FedAvg. In the out-of-distribution scenario (AG News), CoMiGS-1G1S performance tracks the performance of CoMiGS-2G and FedAvg, similar to our observations with the GPT experiments. The complete results are presented in Table 6.

Table 6: Mean test perplexity over the users with homogeneous models, averaged across 3 seeds. Mean (std) for Llama3.2 (1B) model.

|  | In Distribution | Out of Distribution |  |  |
|---|---|---|---|---|
|  | Common-Corpus | AG News | SlimPajama | Multilingual |
| Pretrained | 30.40 | 29.37 | 12.45 | 14.25 |
| Centralized | 17.36 (0.08) | 16.12 (0.05) | 9.58 (0.19) | 11.27 (0.07) |
| Local | 20.19 (0.11) | 19.96 (0.01) | 11.84 (0.06) | 10.93 (0.04) |
| FedAvg | 21.95 (0.11) | 15.86 (0.05) | 11.30 (0.03) | 10.57 (0.05) |
| CoMiGS-2S (ours) | 18.46 (0.13) | 18.03 (0.11) | 11.95 (0.05) | 10.88 (0.03) |
| CoMiGS-2G (ours) | 20.18 (0.09) | 15.41 (0.05) | 11.33 (0.02) | 10.57 (0.03) |
| CoMiGS-1G1S (ours) | 18.37 (0.03) | 16.31 (0.05) | 11.44 (0.02) | 10.60 (0.02) |

## E   ADDITIONAL EXPERIMENTS

We replicate the experiments in Section 4.3 with the SlimPajama dataset, where we assign four times as many tokens to ArXiv User and Book User as to Stack Exchange User and Codes User.

**More Specialists Help with Higher Data Quantity.** From Figure 11, it is evident that ArXiv User and Book User, with abundant local data, benefit from having more local experts.

**Generalists Help to Prevent Redundant Specialists from Over-Fitting?** From Figure 12, we observe more prominent overfitting than in Figure 5, likely because the tasks are objectively easier, as

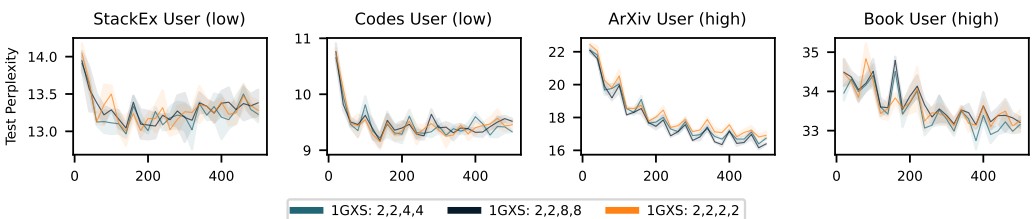

Figure 11: Test Perplexity during training for the SlimPajama setup. ArXiv User and Book User have more local data and thus benefit from having more experts. The numbers in the legend indicate the number of experts $n_i$ within each user. Top-2 routing is performed.

indicated by lower test perplexity from the beginning of fine-tuning. Generalists have limited power to prevent overfitting with easy tasks.

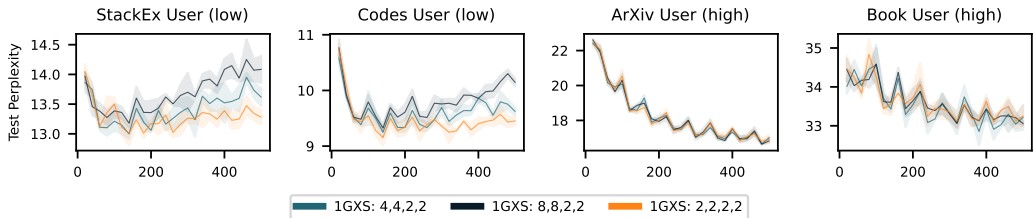

Figure 12: In this SlimPajama setup, Stack Ex User and Codes User despite having low resources locally, overfit slightly on their small-sized local data. Numbers in the legend denote the number of experts $n_i$ within each user. Top2 routing is performed.

**Specialists Can Benefit Generalists.** Low-resourced users that can only support a single expert setup still benefit from collaboration, as the generalist knowledge is refined through a more detailed distinction between specialist and generalist roles via other high-resourced users. This is indicated by the enhanced performances for Stack Exchange and Codes Users.

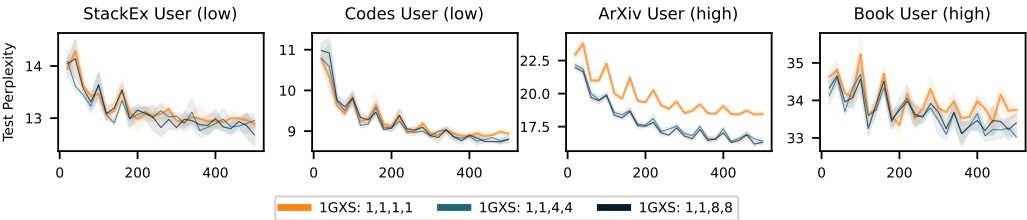

Figure 13: In this SlimPajama setup, Stack Ex User and Codes User, despite having only one expert locally, still benefit from other users having more experts, thereby enhancing the generalist's performance. The numbers in the legend indicate the number of experts, $n_i$, within each user. Top-2 routing is applied when $n_i \geq 2$

## F    VISUALIZATION OF EXPERT SPECIALIZATION

To visualize which tokens are routed to the generalist and specialist experts for our `CoMiGS-1G1S` model trained on SlimPajama, we ask ChatGPT to generate texts in the style of StackExchange, Python Codes, ArXiv Paper and Books. We then feed those texts to the user-specific models and color the token with the Top1 routed index. The routing results after the very first layer (0th), a middle layer (5th), and the very last layer (11th) are presented in Figure 14, 15 and 16.

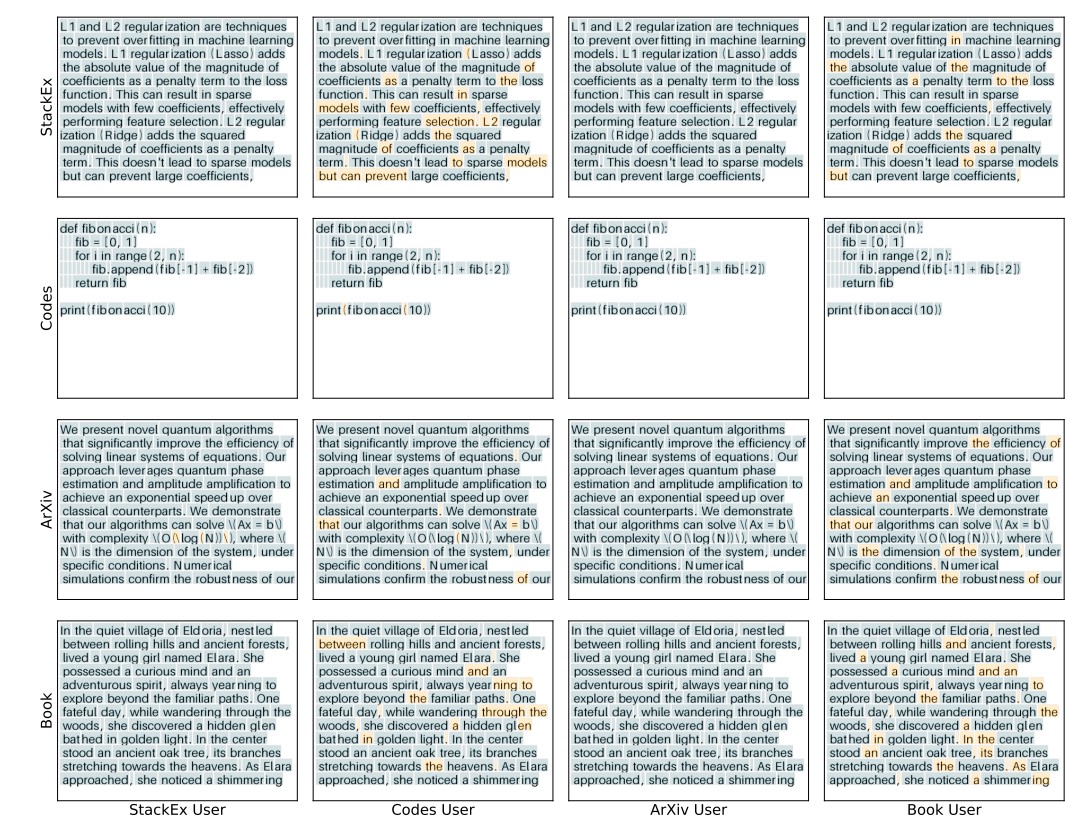

Figure 14: Visualization of token-level routing results for `CoMiGS-1G1S` trained on SlimPajama. Tokens are colored with the first expert choice at the 0th (first) layer. Orange denotes the generalist and blue denotes the specialist. Diagonal entries are in-distribution texts and off-diagonal entries are out-of-distribution texts. Texts are generated by ChatGPT.

We perform the same experiments on AG News, asking ChatGPT to generate News text on the topics World, Sports, Business, and Sci/Tech. The routing results after the very first layer (0th), a middle layer (5th), and the very last layer (11th) are presented in Figure 17, 18 and 19.

For all the plots, diagonal entries are *in-distribution* texts and off-diagonal entries are *out-of-distribution* texts.

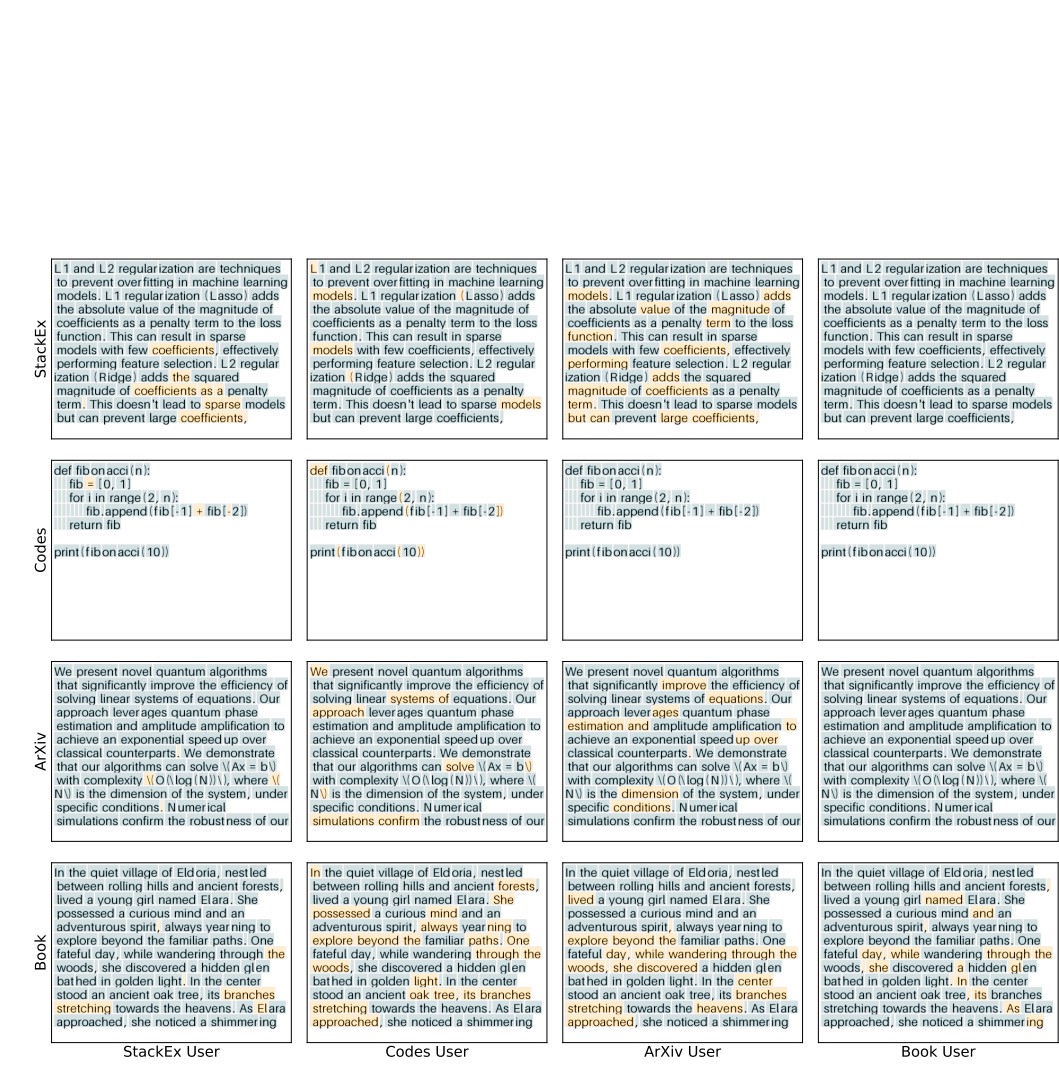

Figure 15: Visualization of token-level routing results for `CoMiGS-1G1S` trained on SlimPajama. Tokens are colored with the first expert choice at the 5th layer. Orange denotes the generalist and blue denotes the specialist. Diagonal entries are in-distribution texts and off-diagonal entries are out-of-distribution texts. Texts are generated by ChatGPT.

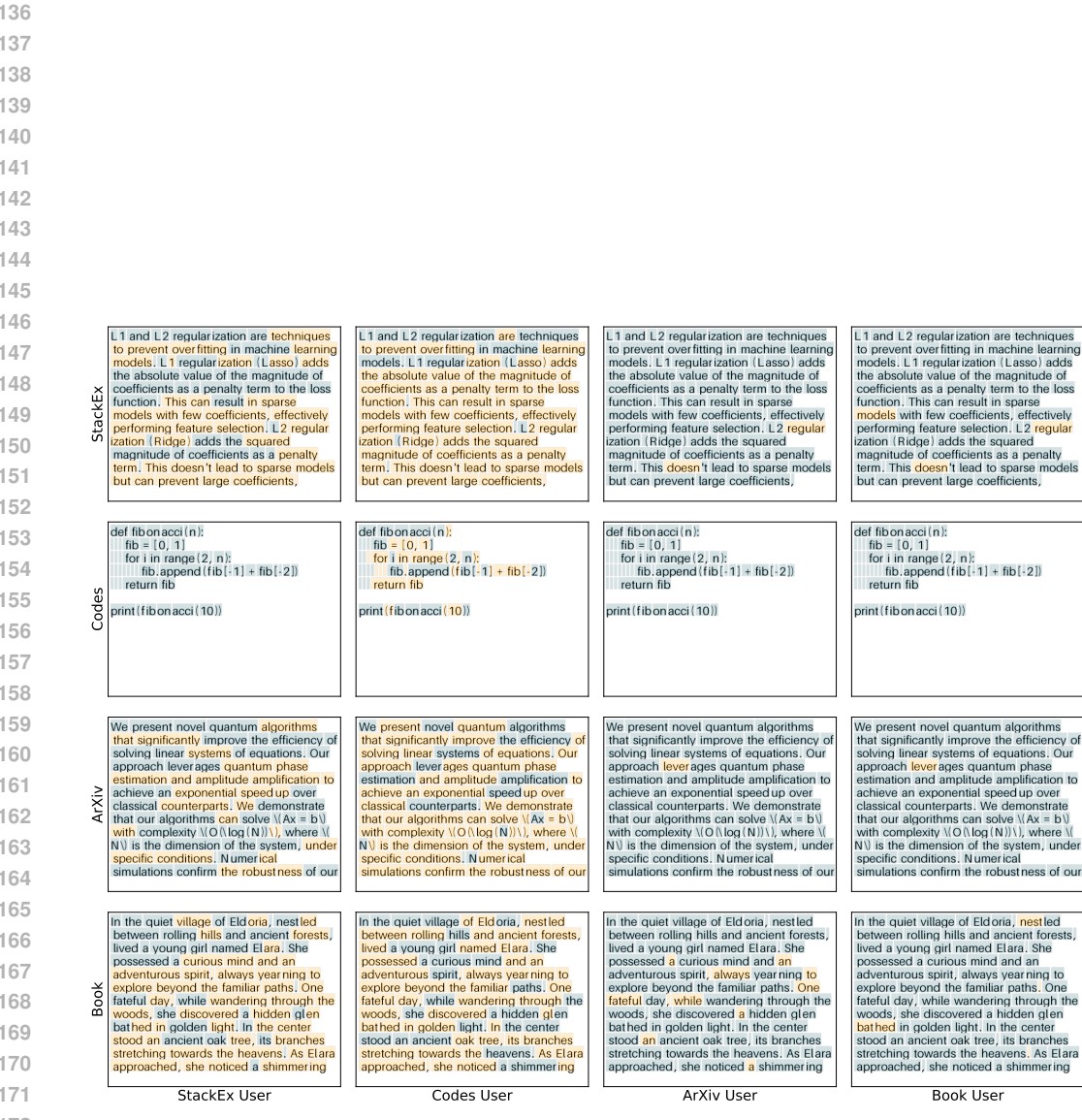

Figure 16: Visualization of token-level routing results for `CoMiGS-1G1S` trained on SlimPajama. Tokens are colored with the first expert choice at the 11th (last) layer. Orange denotes the generalist and blue denotes the specialist. Diagonal entries are in-distribution texts and off-diagonal entries are out-of-distribution texts. Texts are generated by ChatGPT.

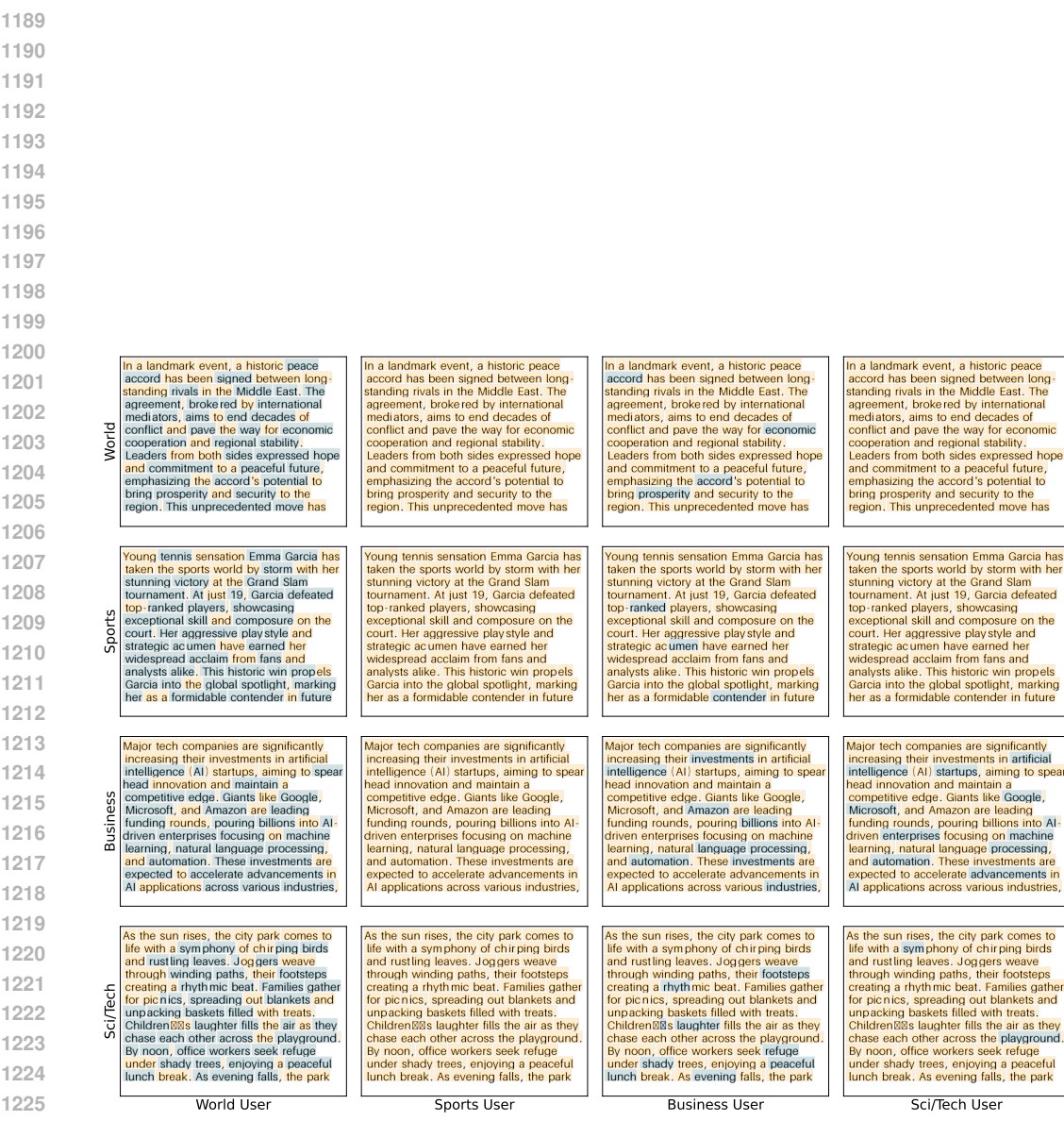

Figure 17: Visualization of token-level routing results for `CoMiGS-1G1S` trained on AG News. Tokens are colored with the first expert choice at the 0th (first) layer. Orange denotes the generalist and blue denotes the specialist. Diagonal entries are in-distribution texts and off-diagonal entries are out-of-distribution texts. Texts are generated by ChatGPT.

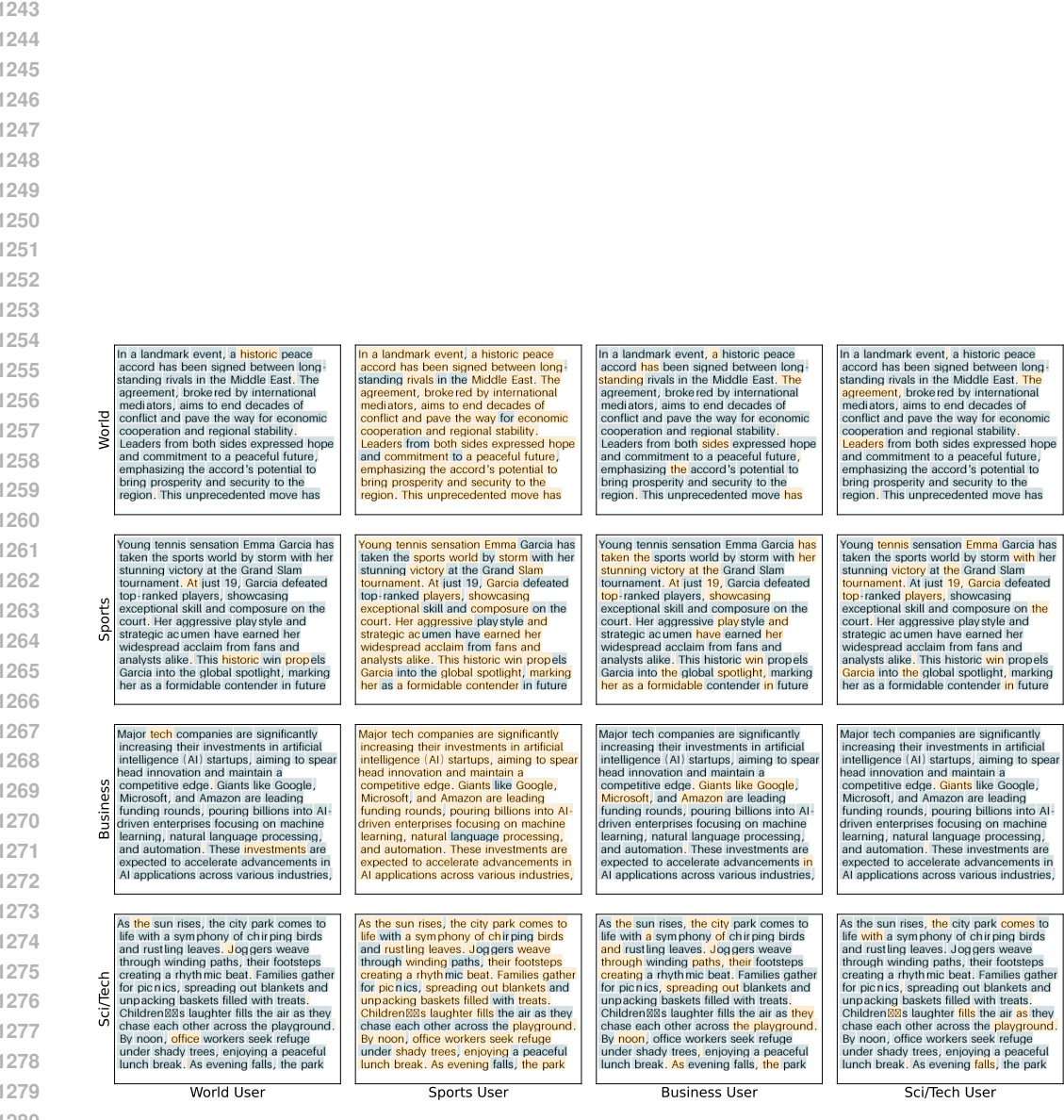

Figure 18: Visualization of token-level routing results for `CoMiGS-1G1S` trained on AG News. Tokens are colored with the first expert choice at the 5th (middle) layer. Orange denotes the generalist and blue denotes the specialist. Diagonal entries are in-distribution texts and off-diagonal entries are out-of-distribution texts. Texts are generated by ChatGPT.

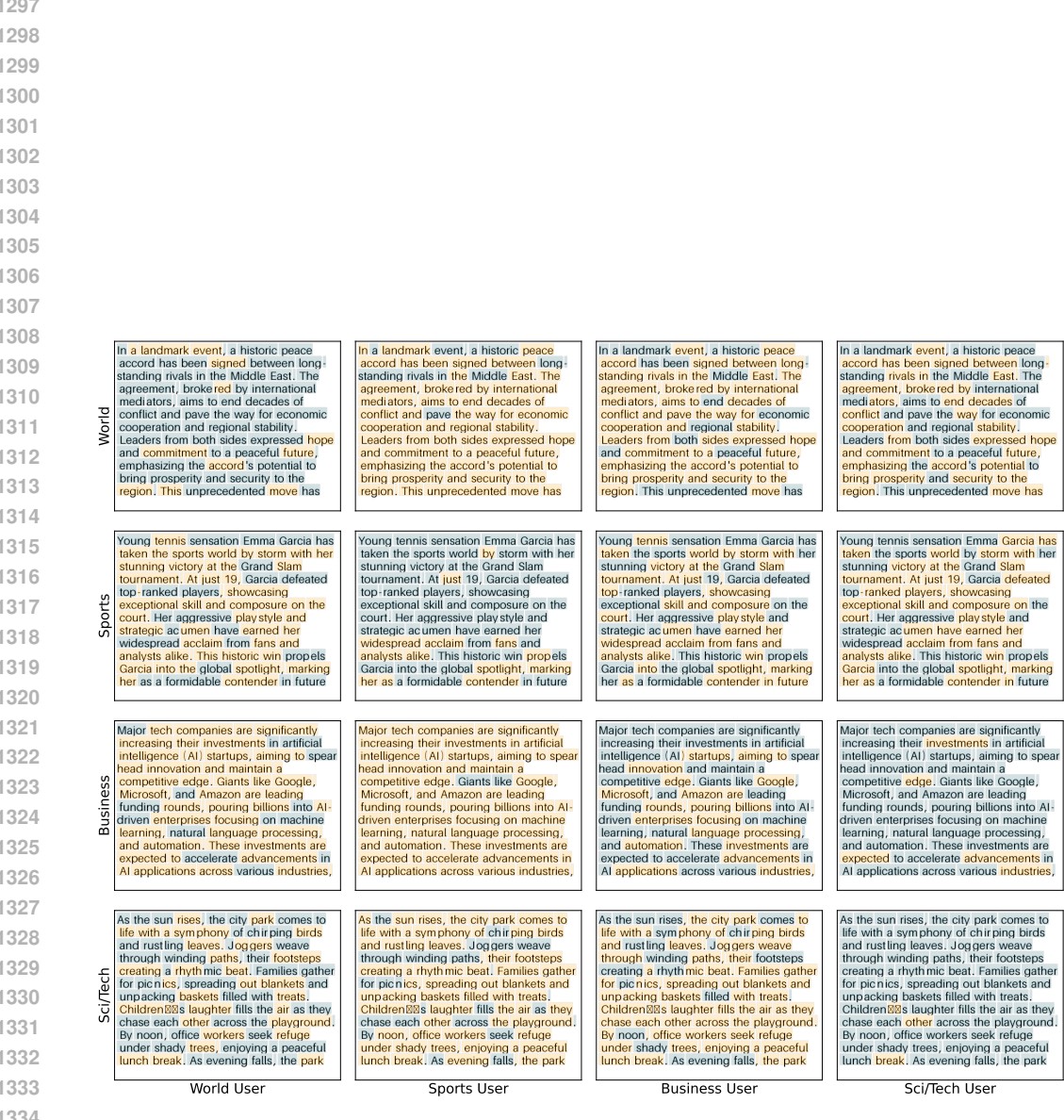

Figure 19: Visualization of token-level routing results for `CoMiGS-1G1S` trained on AG News. Tokens are colored with the first expert choice at the 11th (last) layer. Orange denotes the generalist and blue denotes the specialist. Diagonal entries are in-distribution texts and off-diagonal entries are out-of-distribution texts. Texts are generated by ChatGPT.

## G  ALTERNATING MINIMIZATION CONVERGENCE

### G.1  NOTATION

Let us consider two differentiable functions $f_1(x, y)$ and $f_2(x, y)$, where $x \in \mathbb{R}^d$ and $y \in \mathbb{R}^n$ are some variables. Note that $f_1$ and $f_2$ are simply $\mathcal{L}(f(\boldsymbol{X}_i^{\text{valid}}; \boldsymbol{\Theta}, \phi_i), \boldsymbol{X}_i^{\text{valid}})$ and $\mathcal{L}(f(\boldsymbol{X}_i^{\text{train}}; \boldsymbol{\Theta}, \phi_i), \boldsymbol{X}_i^{\text{train}})$ in our algorithm, and $(x, y)$ are $(\boldsymbol{\Theta}, \phi_i)$.

We are interested to analyze the following *alternating minimization algorithm*, starting from some initial $x_0 \in \mathbb{R}^d$, we denote for every $k \geq 0$:

$$\begin{aligned} y_{k+1} &= \arg\min_{y} f_1(x_k, y), \\ x_{k+1} &= \arg\min_{x} f_2(x, y_{k+1}). \end{aligned} \tag{7}$$

If $f_1 \equiv f_2$ that would be a standard alternation minimization as for minimizing one function $f_1$. However, in our setting $f_1$ and $f_2$ can be different.

For a fixed $x$ and $y$, let us denote the corresponding $\arg\min$ operators by

$$u_1(x) := \arg\min_{y} f_1(x, y)$$

and

$$u_2(y) := \arg\min_{x} f_2(x, y).$$

Using this notation, we can rewrite algorithm equation 7 as follows:

$$y_{k+1} = u_1(x_k), \qquad x_{k+1} = u_2(y_{k+1}), \qquad k \geq 0. \tag{8}$$

Let us further define the following operators, each transforming its own space, for any $x \in \mathbb{R}^d$ and $y \in \mathbb{R}^d$:

$$T(x) := u_2(u_1(x)) \in \mathbb{R}^d,$$

$$P(y) := u_1(u_2(y)) \in \mathbb{R}^n.$$

With this notation, we can rewrite the sequence $\{x_k\}_{k \geq 0}$ simply as

$$x_{k+1} = T(x_k), \qquad k \geq 0. \tag{9}$$

Our **main assumption** on functions $f_1$ and $f_2$ is the following one.

**Assumption 1** *There exist $x^\star \in \mathbb{R}^d$ and $y^\star \in \mathbb{R}^n$ such that*

$$x^\star = T(x^\star) \quad \text{and} \quad y^\star = P(y^\star) \tag{10}$$

**Remark 1** *Note that if $f_1 \equiv f_2 \equiv f$, condition equation 10 holds for the global minimizer of our function $(x^\star, y^\star) = \arg\min_{x,y} f(x, y)$.*

**Remark 2** *It remains an interesting open question: which joint conditions on $f_1$ and $f_2$ imply equation 10.*

### G.2  CONTRACTION AND CONVERGENCE

Depending on a structure of $f_1$ and $f_2$, we might obtain different convergence properties. Let us consider one simple case when the corresponding mappings $u_1$ and $u_2$ are *contractions*, which will imply global linear convergence rates.

We assume the following.

**Assumption 2** *For any fixed $x$ and $y$, let $f_1(x, \cdot)$ and $f_2(\cdot, y)$ be strongly convex with constants $\mu_1, \mu_2 > 0$. Therefore, it holds*

$$f_1(x, y) \geq f_1(x, u_1(x)) + \tfrac{\mu_1}{2}\|y - u_1(x)\|^2, \quad \forall x, y, \tag{11}$$

*and*

$$f_2(x, y) \geq f_2(u_2(y), y) + \tfrac{\mu_2}{2}\|x - u_2(y)\|^2, \quad \forall x, y. \tag{12}$$

Without loss of generality, let us consider the first function $f_1$. We take two arbitrary points $x, \bar{x} \in \mathbb{R}^d$. Applying inequality equation 11 two times, we get

$$f_1(x, u_1(\bar{x})) \geq f_1(x, u_1(x)) + \tfrac{\mu_1}{2}\|u_1(\bar{x}) - u_1(x)\|^2,$$

$$f_1(\bar{x}, u_1(x)) \geq f_1(\bar{x}, u_1(\bar{x})) + \tfrac{\mu_1}{2}\|u_1(x) - u_1(\bar{x})\|^2.$$

Summing up these inequalities, we obtain

$$\mu_1\|u_1(x) - u_1(\bar{x})\|^2 \leq f_1(x, u_1(\bar{x})) - f_1(x, u_1(x)) + f_1(\bar{x}, u_1(x)) - f_1(\bar{x}, u_1(\bar{x})). \tag{13}$$

To proceed with the right hand side, let us assume the following particular structure, that is common to some applications (Nesterov, 2020).

**Assumption 3** *Function $f_1$ has the following representation,*

$$f_1(x, y) \equiv h(x) + g(y) + \langle A(x), B(y) \rangle,$$

*where $h$ and $g$ are convex functions and $A$ and $B$ are Lipschitz operators with constants $L_A$ and $L_B$.*

Using this representation, we can bound the right hand side of equation 13 as follows,

$$\mu_1\|u_1(x) - u_1(\bar{x})\|^2 \leq \langle A(x) - A(\bar{x}), B(u_1(x)) - B(u_1(\bar{x})) \rangle$$

$$\leq \|A(x) - A(\bar{x})\| \cdot \|B(u_1(x)) - B(u_1(\bar{x}))\|$$

$$\leq L_A L_B \|x - \bar{x}\| \cdot \|u_1(x) - u_1(\bar{x})\|.$$

Hence, we obtain the following statement.

**Proposition 1** *Let $\mu > L_A L_B$. Then operator $x \mapsto u_1(x)$ is a contraction:*

$$\|u_1(x) - u_1(\bar{x})\| \leq \tfrac{L_A L_B}{\mu}\|x - \bar{x}\|, \qquad \forall x, \bar{x}. \tag{14}$$

Using this machinery, we see that the following assumption on $u_1$ and $u_2$ can be feasible to achieve.

**Assumption 4** *Let $u_1$ and $u_2$ be contractions with some constants $0 < \lambda_1, \lambda_2 < 1$:*

$$\begin{aligned}
\|u_1(x) - u_1(\bar{x})\| &\leq \lambda_1\|x - \bar{x}\|, &&\forall x, \bar{x} \in \mathbb{R}^d, \\
\|u_2(y) - u_2(\bar{y})\| &\leq \lambda_2\|y - \bar{y}\|, &&\forall y, \bar{y} \in \mathbb{R}^n.
\end{aligned} \tag{15}$$

Under these assumptions we can show the convergence of the sequence $\{x_k\}_{k \geq 0}$ generated by equation 9. Indeed, for every $k \geq 0$, we have

$$\|x_{k+1} - x^\star\| = \|T(x_k) - x^\star\| \overset{equation\ 10}{=} \|T(x_k) - T(x^\star)\|$$

$$= \|u_2(u_1(x_k)) - u_2(u_1(x^\star))\| \overset{equation\ 15}{\leq} \lambda_2\|u_1(x_k) - u_1(x^\star)\|$$

$$\overset{equation\ 15}{\leq} \lambda_1\lambda_2\|x_k - x^\star\|.$$

Therefore, for $k \geq 0$:

$$\|x_k - x^\star\| \leq (\lambda_1\lambda_2)^k\|x_0 - x^\star\|,$$

and we see that $x_k \to x^\star$ with the linear rate. The same reasoning can be applied to the sequence $\{y_k\}_{k \geq 1}$. Thus, we can formally establish the following general convergence result.

**Proposition 2** *Let functions $f_1$ and $f_2$ satisfy Assumption 1 and Assumption 4. Thus the corresponding $\arg\min$ operators $u_1(\cdot)$ and $u_2(\cdot)$ are contractions and their compositions $u_2 \circ u_1$ and $u_2 \circ u_1$ admit fixed points $x^\star$ and $y^\star$ correspondingly. Then, the sequence $(x_k, y_k)_{k \geq 1}$ generated by equation 7 converges to $(x^\star, y^\star)$ with the linear rate.*

