# OpenReview forum: "On-Device Collaborative Language Modeling via a Mixture of Generalists and Specialists"
_ICLR.cc/2025/Conference — Submitted to ICLR 2025_

### Official Review · Reviewer_3sa7 · 2024-11-01

**Soundness:** 2
**Presentation:** 2
**Contribution:** 1
**Rating:** 5
**Confidence:** 5

**Summary:**

This paper introduces a novel collaborative learning framework called CoMiGS (Collaborative learning with a Mixture of Generalists and Specialists) for on-device fine-tuning of Large Language Models (LLMs). CoMiGS addresses both system heterogeneity (differences in local computational resources) and data heterogeneity (differences in user-specific data) by differentiating between generalists, which handle shared knowledge across devices, and specialists, which adapt to user-specific data. The approach is formulated as a bi-level optimization problem, with generalists and specialists iteratively updated through a routing mechanism, which adapts based on the target distribution. Experimental results demonstrate that CoMiGS outperforms existing methods in scenarios with varied data and system settings.

**Strengths:**

- The token-level routing enables CoMiGS to handle data heterogeneity more efficiently than client-level methods, and the bi-level optimization approach aligns well with the personalized and collaborative goals of the framework.
- The paper provides robust experimental evaluations on three datasets, covering both in-distribution and out-of-distribution scenarios.

**Weaknesses:**

- While the paper presents CoMiGS as a novel collaborative learning framework, the core idea of combining generalist and specialist models to handle heterogeneous data is not particularly new. Similar approaches have been proposed in previous federated learning and mixture-of-experts frameworks. Many existing works utilize a combination of shared global models (analogous to generalists) and personalized models (specialists) to handle data and system heterogeneity [1]. The authors do not provide a clear explanation of how CoMiGS fundamentally differs from or improves upon these prior methods, which weakens the claim of novelty.
- The bi-level optimization approach may introduce significant computational overhead, especially for on-device applications with limited resources. Each optimization level requires iterative updates, making the entire process computationally intensive.
- The paper does not propose alternative mechanisms for model evaluation or adaptation that avoid the use of a validation set. Other federated learning approaches often incorporate online learning or lightweight evaluation methods that do not require a dedicated validation split, which CoMiGS could consider.
- The rationale behind maintaining both generalists and specialists within the framework is not entirely convincing. If specialist models are designed to adapt to user-specific data, the role of generalists becomes ambiguous, especially in highly heterogeneous settings where each user's data diverges significantly from the global average. The coexistence of generalists and specialists adds complexity to the model without a clear demonstration of the complementary benefits of this setup.


[1]: FedCP: Separating Feature Information for Personalized Federated Learning via Conditional Policy. Jianqing Zhang, Yang Hua, Hao Wang, Tao Song, Zhengui Xue, Ruhui Ma, Haibing Guan
、

**Questions:**

- Could the authors clarify how CoMiGS fundamentally differs from existing frameworks that use a mixture of shared (global) and personalized (local) models to handle heterogeneity in federated learning? Specifically, what are the unique contributions of CoMiGS in comparison to these prior methods?
- CoMiGS relies on a validation set for tuning and routing adjustments, which may not be feasible in real-world on-device settings due to data limitations. How would CoMiGS function without a dedicated validation set, or are there alternative mechanisms for evaluation that do not require a separate validation split?

---

> ### Author Response · Authors · 2024-11-20
>
> Thank you for your feedback. It appears that some aspects of the mentioned weaknesses may have been misunderstood. We address these points below. If our clarifications resolve some of your concerns, we kindly request you to reconsider the score provided.
>
> W1: We have never claimed in the write-up that our contribution is the proposal of a mixture of shared and personalized models. We refer to the first point of the **public comment** for a detailed clarification of the novelty.
>
> Thanks for pointing FedCP out. We indeed missed this work and will include a discussion in our future iteration. While FedCP indeed has a global and local feature extractor and a conditional policy network (CPN) that acts as a router, our CoMiGS method differs from it in the following aspects: 1) **communication units**: CoMiGS only requires clients to share global experts, while FedCP requires sharing of personalized feature extractors and the CPN (router). CoMiGS is more private and communication efficient. 2) **tackling out-of-distribution tasks**: FedCP updates the router and feature extractors both on the same loss, while our CoMiGS updates the router via a separate validation loss, making the method more robust and able to tackle out-of-distribution tasks. 3) **tackling system heterogeneity**: according to our understanding, FedCP only addresses data heterogeneity, not system heterogeneity. As FedCP initializes personal feature extractors with the global parameters, this requires all personal extractors to share the same structure.
>
> W2: Our bi-level formulation does not add much computation and memory overhead. It is worth noticing that we solve the bi-level optimization via an approximate iterative updating algorithm, which removes the calculation of Hessian and thus the router update only requires first-order gradients over the router, which is a small one-layer MLP (of size 768*2 in our setup). CoMiGS even has less computational complexity than FedCP, as FedCP updates the router each iteration with the feature extractors, while CoMiGS updates the router every 30 iterations, see Appendix B. A detailed numerical analysis regarding computational complexity can be found in the second point of the **public comment**.
>
> W3: We apologize for not making it clearer in the main texts. We indeed pointed out that the validation set can alternatively be replaced by a newly sampled batch from the training set (Line 837-838). For in-distribution tasks, this achieved decent performance. However, for out-of-distribution tasks, this would not necessarily work.
>
> W4: We would like to draw your attention to Table 1 in our manuscript, where we investigate the necessity of having both generalists and specialists through an ablation study. Specifically, CoMiGS-2G and CoMiGS-2S are variants that only allow two generalists or two specialists, respectively. The empirical evidence demonstrates that neither approach consistently performs well across different task scenarios.
> You are correct that when tasks are more homogeneous, generalists suffice, while for highly heterogeneous tasks, specialists excel. However, in practice, it is often impossible to predict task heterogeneity in advance to determine the most suitable method. This is where our CoMiGS method stands out, as it closely tracks the best-performing approach in each scenario, thanks to its token-dependent routing mechanism.

---

> > ### Author Response · Authors · 2024-11-20
> >
> > Q1: _First of all_, we admit that there is a lot of work on a mixture of global and local modules in personalized federated learning. However, they primarily focus on classification-based vision tasks. Balancing personalization and collaboration is a new and non-trivial challenge in the language domain, due to the unique nature of language: even two documents about two drastically different topics still share a lot of common tokens, so that quantification of statistical heterogeneity is already a non-trivial task. In our experimental setup, we always assign one distinct category to one client and make sure the clients have no overlapping categories. Even like this, when a category represents a topic, personalization is preferred, while when a category represents a language, collaboration is preferred. This phenomenon has also been observed in [1], which is fundamentally different than in traditional classification tasks. Our work acts as a pilot study into leveraging collaboration for personalization purposes in the language domain. _Second_, our bi-level formulation of MoE is novel, as the past works [2,3]  update router and expert parameters at the same time, which forbids the adaptation to out-of-distribution tasks. _Third_, according to our best knowledge, no prior works have addressed both system and data heterogeneity in collaborative language modeling, while our work is the first to achieve both.
> >
> > Q2: Our CoMiGS method can work even when a validation set is not available, alternatively, one could sample a new batch to update the router. We would like to draw your attention to Table 4 _iii) Alternating update routing params on newly sampled batches from the training set_, where CoMiGS can still offer decent performance. However, it is worth pointing out that this can only work with in-distribution scenarios, for a target distribution that is different than the training distribution, a small validation set will be needed, as in [4].
> >
> > [1] Personalized Collaborative Fine-Tuning for On-Device Large Language Models. Wagner et al. COLM 2024
> >
> > [2] FedCP: Separating Feature Information for Personalized Federated Learning via Conditional Policy. Zhang et al. KDD 2023
> >
> > [3] pFedMoE: Data-Level Personalization with Mixture of Experts for Model-Heterogeneous Personalized Federated Learning. Yi et al. 2024
> >
> > [4] Personalized Federated Learning with First Order Model Optimization. Michael Zhang et al. ICLR 2021

---

> > > ### Comment · Reviewer_3sa7 · 2024-11-25
> > >
> > > Thank you for your detailed rebuttal and the clarifications provided. I have carefully reviewed your responses and the ensuing discussions. While I acknowledge the efforts made to address the concerns raised, I still find that the methodological novelty is somewhat limited. Consequently, I believe the work does not fully meet the high standards expected for acceptance at ICLR. Therefore, I am inclined to raise my original assessment but below the acceptance threshold.

---

> ### Author Response · Authors · 2024-11-26
>
> Dear Reviewer 3sa7,
>
> Thanks for acknowledging the fact that we managed to address your concerns. Regarding the methodological novelty part, we would like to remind you kindly that not all published works from ICLR have to offer groundbreaking novel methods, for example, FFALoRA [1] is in the same lieu and accepted by ICLR 2024, which suggested a way to improve LoRA in the context of privacy-preserving, instead of proposing a novel method to preserve privacy.
>
> Our work indeed offers novel contributions in other aspects: 1) we are the _first_ to address both data and system heterogeneity in collaborative LLMs with marginal compute overhead over the standard FedAvg; 2) our bi-level formulation of the MoE learning objective is novel; 3) our method brings clear empirical gains with a theoretical convergence guarantee (see Theorem 3.1 in our modified manuscript).
>
> Thank you for your time in advance!
>
> Best,
>
> Submission 3571 authors
>
> [1] Sun et al. Improving LoRA in Privacy-preserving Federated Learning, ICLR 2024

---

### Official Review · Reviewer_RkHW · 2024-11-03

**Soundness:** 3
**Presentation:** 3
**Contribution:** 2
**Rating:** 6
**Confidence:** 4

**Summary:**

This paper presents CoMiGS, a collaborative learning framework for personalized fine-tuning of LLMs on edge devices, addressing data heterogeneity and privacy challenges. CoMiGS features a dual-expert structure with generalists and specialists for effective knowledge sharing while preserving local data privacy. It uses a bi-level optimization mechanism, updating router parameters based on validation loss (outer optimization) and expert parameters based on training loss (inner optimization). This approach mitigates issues seen in traditional MoE architectures. Overall, CoMiGS effectively manages system and data heterogeneity in federated LLM training, enabling efficient data sharing without compromising privacy and enhancing model robustness.

**Strengths:**

1. CoMiGS effectively addresses heterogeneity by integrating knowledge across diverse end users while tackling the challenges of data heterogeneity. Its bi-level optimization mechanism strategically utilizes the roles of generalists and specialists, facilitating both data sharing and personalized experiences for end users.

2. CoMiGS employs load-balancing techniques to ensure a balanced allocation among different experts, enhancing the model’s robustness and generalization capabilities by maintaining equilibrium across them.

**Weaknesses:**

1. CoMiGS resembles a fusion of LoRAMoE [1,2,3,4] and FL-LoRA [5,6]. However, it lacks analytical comparisons with related studies and does not provide experimental comparisons to support its claims.

2. CoMiGS is designed for edge deployment, but fine-tuning multiple LoRA modules (generalists vs. specialists) may incur more computational overhead.

3.  The title suggests a focus on LLMs, but evaluation is limited to GPT-2 (124M), which is too simplistic in terms of both model type and parameter count. The study omits more complex LLMs, such as Llama3-1B, Llama3-3B, and Gemma-2B, which are designed for mobile environments. With the increase in parameters, issues such as embedded knowledge may become more pronounced, potentially enhancing the robustness and adaptability for In-Distribution and Out-of-Distribution task scenarios. This prompts the question: could this technique scale effectively with future generations of LLMs? Further research is necessary to explore these possibilities.

References:

[1] When MOE Meets LLMs: Parameter Efficient Fine-tuning for Multi-task Medical Applications, SIGIR 2024.

[2] Mixture of LoRA Experts, ICLR 2024.

[3] Lorahub: Efficient cross-task generalization via dynamic lora composition, COLM 2024.

[4] Pushing mixture of experts to the limit: Extremely parameter efficient moe for instruction tuning, ICLR 2024.

[5] pFedLoRA: Model-Heterogeneous Personalized Federated Learning with LoRA Tuning, Arxiv 2023.

[6] FDLoRA: Personalized Federated Learning of Large Language Model via Dual LoRA Tuning, Arxiv 2024.

**Questions:**

1. To facilitate both data sharing and end-user personalization within the inherent leader-follower structure, the proposed approach seems to introduce additional computational and communication costs during training and data sharing. Could the authors clarify the impact of this optimization on computational efficiency and memory storage requirements? Additionally, how does this affect the feasibility of deploying CoMiGS on resource-constrained edge devices?

---

> ### Author Response · Authors · 2024-11-20
>
> Thank you for your critical feedback, we address your concerns and questions as follows:
>
> W1: Yes, there is a growing body of work focusing on using LoRA MoE for multi-task learning. We will incorporate these works into the related works section in future iterations. Additionally, we reviewed [5] and [6] as referenced by you. [5] is a study in the vision domain. [6] is a concurrent work that shares similarities with our CoMiGS method, which we missed during our literature search. Thanks for pointing it out. While FDLoRA also incorporates global and local LoRA modules alongside a learnable set of weights to combine them, there are significant differences from our approach:
>
> **Training Strategy**: FDLoRA trains global and local experts sequentially, where local experts are initialized from global ones, causing the local experts to remain close to the global experts. In contrast, our CoMiGS method trains these experts in parallel, enabling them to co-adapt dynamically, which is supposed to lead to a greater distinction between the global and local experts.
>
> **Weight Granularity**: FDLoRA employs client-wise learnable weights, whereas our method takes a more fine-grained approach by learning token-wise weights.
>
> Since the implementation of [6] is unavailable, we recreated their method in our codebase using the hyperparameters provided in their paper.
> The results of this implementation are presented in the table below: FDLoRA failed to outperform our CoMiGS-1G1S. We noticed that FDLoRA performs similarly to CoMiGS-2G method, where we assign two generalists per client. This indicates that indeed the local and global experts trained from FDLoRA are quite close.
>
> We also highlight a key limitation of FDLoRA: it does not support different numbers of experts across clients. This is a crucial feature in our method, as it effectively addresses system heterogeneity, allowing for greater flexibility in diverse client setups.
>
> || MultiLingual | SlimPajama| AG News|
> |------------|------------|------------|------------|
> | FDLoRA | 57.45 (0.81) | 22.71 (0.40) | 33.61(0.07) |
> | CoMiGS - 2S (ours) | 46.36 (0.16) | 22.51 (0.08) | 35.81 (0.13) |
> | CoMiGS - 2G (ours) | 58.31 (0.17) | 21.36 (0.01) | 31.18 (0.05) |
> | CoMiGS - 1G1S (ours) | 47.19 (0.10) | 21.79 (0.04) | 33.53 (0.03) |
>
>
> W2: We would like to clarify that our method does not imply that devices should have multiple LoRA modules. Instead, the number of LoRA modules per device can vary depending on the edge device capacity. Those resource-limited edge devices can operate effectively using only the generalist expert. As demonstrated in Figure 6, edge devices equipped with just one generalist expert can still benefit from collaboration with more powerful devices: the first device achieves better performances when the numbers of LoRA modules across the clients change from [1,1,1,1] to [1,1,8,8]. To the best of our knowledge, our proposed method is the first to address both system heterogeneity and data heterogeneity simultaneously.
>
> W3: Thanks for pointing this out. We understand your concern over the scaling ability of our method to larger and more recent LLMs. However,  scaling up to larger models becomes less practical within an academic budget, as we even need to create multiple clients at one time, which is extensive in memory and thus hinders us from using larger models. So far, we can fit all our experiments on an A100 GPU.
>
> We are currently working on a LLAMA 3.2 - 1B base model, however, the proper comparison between the baselines requires a bit more time. We will follow up with the results in the following days.
>
> Q1: Please refer to the second point of our **public comment** for a detailed numerical comparison of the computational/communication overhead and memory requirements. Our CoMiGS method is rather resource-efficient.
> As we previously stated in W2, our framework can easily cope with resource-constrained devices. Furthermore, they can benefit from collaboration with resource-rich devices.

---

> ### Author Response · Authors · 2024-11-24
> **Update on the LLAMA results**
>
> We're pleased to share the results of our LLAMA 3.2-1B base model comparison, following up on our previous response.
>
> Given the extensive pre-training of LLAMA 3 models on over 15 trillion tokens from public sources [2], and the multilingual capabilities of LLAMA 3.2 - 1B [1], fine-tuning on multilingual Wikipedia or SlimPajama resulted in negligible improvements (see Table below) likely due to significant overlap with the pre-training data corpus.
>
> To clarify the differences between the methods, we propose a new fine-tuning dataset, which is derived from Common Corpus (specifically, the YouTube-Commons, Latin-PD, and TEDEUTenders collections) and the Harvard USPTO dataset. Following our previous methodology, each client is assigned one of the datasets to maximize heterogeneity. We use this dataset to model the in-distribution scenario. Additionally, we reduced the number of training iterations for the Agnews experiment.
>
> Our results on the Common Corpus-based dataset, which emphasizes domain-specific language and structure, demonstrate that our CoMiGS-1G1S method can outperform local training and FedAvg. In the out-of-distribution scenario (Agnews),  CoMiGS-1G1S performance tracks the performance of CoMiGS-2G and FedAvg, similar to our observations with the GPT experiments. The complete results are presented below.
>
> |             | Common-Corpus | Agnews     | SlimPajama   | Multilingual |
> | ----------- | --------------| ------------------| ------------ | ------------ |
> | Pretrained    | 30.40         | 29.37             | 12.45        | 14.25        |
> | Centralized | 17.36 (0.08)  | 16.12 (0.05)      | 9.58 (0.19)  | 11.27 (0.07) |
> | Local       | 20.19 (0.11)  | 19.96 (0.01)      | 11.84 (0.06) | 10.93 (0.04) |
> | Fedavg      | 21.95 (0.11)  | 15.86 (0.05)      | 11.30 (0.03) | 10.57 (0.05) |
> | CoMiGS-2S   | 18.46 (0.13)  | 18.03 (0.11)      | 11.95 (0.05) | 10.88 (0.03) |
> | CoMiGS-2G   | 20.18 (0.09)  | 15.41 (0.05)      | 11.33 (0.02) | 10.57 (0.03) |
> | CoMiGS-1G1S | 18.37 (0.03)  | 16.31 (0.05)      | 11.44 (0.02) | 10.60 (0.02) |
>
> [1] https://ai.meta.com/blog/llama-3-2-connect-2024-vision-edge-mobile-devices/
>
> [2] https://ai.meta.com/blog/meta-llama-3/

---

### Official Review · Reviewer_6KmX · 2024-11-04

**Soundness:** 2
**Presentation:** 3
**Contribution:** 2
**Rating:** 5
**Confidence:** 3

**Summary:**

The article introduces CoMiGS, a collaborative learning approach designed to address the challenges of system and data heterogeneity in on-device large language models (LLMs). By combining generalists and specialists, CoMiGS facilitates knowledge sharing while also personalizing solutions for individual user datasets. A key innovation of this approach is its bi-level optimization framework for the Mixture-of-Experts (MoE) objective, which allows for flexible aggregation and adaptation to the varying computational resources of users. CoMiGS demonstrates improved performance over existing methods, such as pFedMoE, by enhancing routing mechanisms and accommodating a flexible number of experts for each user. This ultimately leads to better privacy and personalization in federated learning.

**Strengths:**

- The article exhibits clear writing and a well-organized structure, effectively distinguishing its contributions from existing works and facilitating reader comprehension.
- The article addresses the potential applications of CoMiGS within the realms of federated learning and large language models, highlighting the method's broad applicability and promising implications.

**Weaknesses:**

1. The study is based on a limited scale, involving only four clients. It would be beneficial to explore methods to further differentiate between clients, such as employing clients with the same domain data but varying quantities.

2. While the article emphasizes the relationship between large language models (LLMs) and personalized federated learning, the novelty appears insufficient. The approach seems to replace traditional personalization methods in federated learning with LoRA training for LLMs and uses the Mixture-of-Experts (MoE) framework for personalizing each LoRA module. The motivation section needs to reinforce the necessity of this approach to establish its distinct contribution.

3. The article does not adequately address how it resolves issues related to resource heterogeneity. The experimental validation in this area appears insufficient, particularly regarding the fixed nature of experts for each client, as client resources may fluctuate over time.

**Questions:**

see weakness

---

> ### Author Response · Authors · 2024-11-20
>
> Thanks for your feedback. We address your raised concerns and questions as follows:
>
> W1: In our write-up, we selected the most challenging scenario for collaboration, where each user is assigned a distinct, non-overlapping category. While this setup emphasizes the need for personalization, our results demonstrate that collaboration remains crucial and offers significant benefits.
>
> In response to your suggestion, we conducted additional experiments using homogeneous data. Specifically, we assigned all four clients French Wikipedia texts containing 100,000, 500,000, 1,000,000, and 5,000,000 tokens each. In this scenario, we expected FedAvg to perform best, which was indeed observed. For local training, users with smaller datasets experienced overfitting. However, our CoMiGS method effectively leveraged the generalist expert to mitigate overfitting, demonstrating its effectiveness in homogeneous data settings as well.
>
>
> || FedAvg | Local | CoMiGS|
> |------------|------------|------------|------------|
> | | 34.93 (0.29) | 68.92 (1.67)| 37.33 (0.35) |
>
> W2: We agree that the combination of MoE and LoRA is not novel, and we have not claimed any originality regarding them. However, we would like to emphasize that the novelty of our work lies in: (1) the bi-level formulation of the MoE objective, which allows seamless handling of out-of-distribution tasks, (2) being the first to address both system and data heterogeneity in collaborative LLMs, and (3) effectively disentangling resource abundance from data quantity.
> We consider these significant contributions to advancing collaborative LLMs.
>
> Furthermore, our work highlights an interesting phenomenon: the heterogeneity of language cannot be quantified in the same way as in the vision domain. While all clients are consistently assigned distinct categories (e.g., different topics or languages), the preferences over local training or FedAvg may vary, as shown in Table 1. This is because language, unlike visual data, inherently consists of numerous repeating semantic units, leading to unique dynamics in how heterogeneity manifests and is addressed. We will make this more explicit as well in our future iteration.
>
> We also refer you to the first point of our **public comment** for a detailed clarification on the novelty/contribution of our method.
>
>
> W3: Our discussion of resource heterogeneity primarily focuses on the _device_ axis rather than the _time_ axis, as detailed in Section 4.3. We demonstrated that resource heterogeneity can be effectively addressed by enabling devices to utilize varying numbers of specialists. Regarding your concern about potential fluctuations in device resources over time, such scenarios may indeed emerge with advancements in edge devices, such as mobile phones. Expanding existing pretrained experts by adding new ones is a research area in its own right. We refer to [1] for an example, of which the technique can be complementary to our method.
>
> [1] MoExtend: Tuning New Experts for Modality and Task Extension. Zhong et al.

---

> > ### Comment · Reviewer_6KmX · 2024-11-26
> >
> > Thank you for your detailed response and for conducting additional experiments to address our concerns. I appreciate the effort you put into clarifying the novelty of your work and expanding the evaluation with homogeneous data scenarios. The added experiments indeed provide further insights into the effectiveness of CoMiGS, and I acknowledge the demonstrated improvements in mitigating overfitting.
> >
> > However, I remain unconvinced regarding the claimed contributions related to resource heterogeneity. Your response highlights a focus on the **device axis** rather than the **time axis**. Even within the scope of the device perspective, I believe the treatment of resource heterogeneity in your method is not sufficiently robust. In federated learning (FL), the set of devices participating in each round is highly dynamic. This inherent device-level dynamism introduces unique challenges that go beyond what is addressed in your current formulation. Specifically, I would like to raise the following points for clarification:
> >
> > 1. **Limitations of Expert Updates and Routing Parameters:** In scenarios with dynamic device participation, how do your proposed updates of expert models and routing parameters effectively handle such variability? Could you elaborate on how your method remains effective under the practical constraint of devices joining or leaving during the learning process?
> >
> > 2. **Nature of Resource Heterogeneity:** Your response does not sufficiently clarify whether the observed resource heterogeneity stems from **computation heterogeneity**, **communication heterogeneity**, or both. These two aspects often require distinct handling strategies, and it is not clear whether a personalization-based approach (e.g., LoRA + MoE) can universally address them.
> >
> > 3. **Need for Specificity:** Resource heterogeneity in FL cannot be reduced to a general concept. While personalized solutions like LoRA and MoE are promising, their application requires a more nuanced understanding of the heterogeneity in question. Without this, the proposed method risks being too generalized for practical deployment.
> >
> > Given these concerns, I believe that your current handling of resource heterogeneity requires further clarification and refinement. While your method demonstrates promise, its limitations in addressing the dynamic nature of device participation in FL need to be explicitly acknowledged and addressed in future iterations of the work. As such, I will maintain my current score.

---

> > > ### Author Response · Authors · 2024-11-26
> > >
> > > Dear Reviewer 6KmX,
> > >
> > >
> > > Thanks for your response. Here are our clarifications:
> > >
> > >
> > > 1) Regarding dynamic device participation, our method should remain effective as long as the global expert parameters can be aggregated. If a device joins, it simply downloads the shared global expert parameters and begins training according to our algorithm. If a device leaves, it can seamlessly transition to standard MoE training.
> > >
> > >
> > > 2) In our context, our system heterogeneity solely refers to varying computational resources across devices (see lines 023-024 in the abstract of our manuscript:  **Our approach accommodates users’ varying computational resources through different numbers of specialists**). We consider the same setup for system heterogeneity as in FlexLoRA [1] and HetLoRA [2], which is standard in the field. We did not consider communication heterogeneity in this work.
> > >
> > >
> > > 3) As demonstrated by our experiments, our method consistently delivers strong performance regardless of the presence or absence of data heterogeneity. This was simulated by assigning either different or identical data categories to different users. We have clear evidence that our method is robust to varying levels of data heterogeneity.
> > >
> > >
> > > We acknowledge that the dynamic nature of device participation in FL is an important practical concern that we did not focus on in this work. However, we emphasize that our contributions address significant challenges related to **data heterogeneity and computational heterogeneity**, which remain valuable in practical scenarios.
> > >
> > >
> > > With respect to your comment that “the dynamic nature of device participation in FL need to be explicitly acknowledged and addressed in future iterations of the work”, we would greatly appreciate any references to other works that address dynamic device participation, data heterogeneity, communication, and computation heterogeneity. These would help us explore them and incorporate insights or compare our approach to these works.
> > >
> > >
> > > Best,
> > >
> > > Submission 3571 Authors
> > >
> > > [1] Federated Fine-tuning of Large Language Models under Heterogeneous Tasks and Client Resources. Neurips 2024
> > >
> > > [2] Heterogeneous LoRA for Federated Fine-tuning of On-Device Foundation Models. EMNLP 2024

---

> ### Author Response · Authors · 2024-11-25
>
> Dear Reviewer 6KmX
>
>
> Thank you for the time and effort you have dedicated to reviewing our manuscript and for providing thoughtful feedback.
>
>
> In response to your suggestions, we have added a new experiment exploring different data quantities, which demonstrates the consistent performance of our method. Additionally, we have updated the manuscript to include a convergence proof and a new base model (Llama3.2 - 1B). These additions further support the theoretical and empirical soundness of our proposed method.
>
>
> Given these improvements, we kindly request a re-evaluation of our work. We understand this is a particularly busy period, and we greatly appreciate your time and consideration.
>
>
> Thank you,
>
>
> Submission 3571 Authors

---

### Official Review · Reviewer_HA1D · 2024-11-10

**Soundness:** 3
**Presentation:** 4
**Contribution:** 2
**Rating:** 6
**Confidence:** 3

**Summary:**

This work introduces an approach for collaborative, on-device language modeling based on Federated Learning. It addresses the challenges of system and data heterogeneity, common in federated learning configurations. The focus is on language modeling tasks, specifically MoE-based LLMs using LoRA. The novelty lies in a bi-level optimization formulation for the MoE learning objective, which effectively balances collaboration and personalization. This approach is particularly advantageous in scenarios with high data heterogeneity across multiple datasets and use cases.

**Strengths:**

- Well-written paper. Nicely structured, well motivated, and easy to follow. Overall, excellent presentation.
- I appreciate the effort to publish the code, together with the submission, while preserving their anonymity. This is not always the case. In addition, the code is of high quality.
- While the approach that the authors have followed is simple, it is highly practical and demonstrates an effective balance when addressing data and system heterogeneity.
- Carefully chosen baselines to compare the performance.

**Weaknesses:**

- My biggest concern is regarding the significance of its contribution, especially considering the heavy overlap with the listed related work "pFedMoE: Data-Level Personalization with Mixture of Experts for Model-Heterogeneous Personalized Federated Learning". I understand that the authors have clearly and fairly mentioned the differences. However, I can't deny that the other work is stealing some of its contributions.
- I would have expected a theoretical analysis section in such a paper. Instead, all results and discussions are derived from simulations.
- While I said the manuscript is very well written, I do think it is missing some background work that would make it easier to follow. For instance, a few sentences about what MoE is, what the router is, and its role, what is LoRA, etc. I am aware of the space limit, but I felt that this was missing from the paper.
- Similar to the previous comment, it is missing some (more generic) relevant related works on the topic of addressing system and data heterogeneity during FL configurations. I understand that this work is focused on language modeling tasks; however, I believe the authors should mention (at least briefly in 1-2 lines) a few examples in more classical DNN (non-LLM) scenarios. Some examples:
  * Fjord: Fair and accurate federated learning under heterogeneous targets with ordered dropout (Horvath et al.)
  * Revisiting Sparsity Hunting in Federated Learning: Why does Sparsity Consensus Matter? (Babakniya et al.)
- The authors mention privacy as the motivation of using on-device LLMs (that's fair for sure), but also when applying FL. There are numerous attacks on the FL space that render such claims unrealistic, and perhaps such claims should be phrased more cautiously.

**Questions:**

- Is there any overhead (communication, computation) compared to classical FL approaches? Especially considering the alternating update of theta and fi.
- Is there a particular reason that a theoretical analysis is missing from this paper?
- On the privacy aspect mentioned above, do you think that privacy risk changes depending on the device capabilities? E.g., more capable devices (higher capacity) will aggregate more information, and thus be more exposed?

---

> ### Author Response · Authors · 2024-11-20
>
> Thank you for your encouraging feedback! Regarding your raised concerns, we address them as follows:
>
>
> W1: Please refer to the first point of our **public comment**.
>
>
> W2: We refer to the existing theoretical analysis of alternating minimization provided,e.g., in [1] (see their Theorem 1) or [2], which can be directly applied to our algorithm under the condition that $f(X^{\text{train}}, \cdot)$ and $f(X^{\text{valid}}, \cdot)$ are sufficiently close. This serves as the primary motivation for developing this algorithm.
>
>
> We checked the convergence guarantee of our proposed algorithm in the past days. Assume that there exists a unique global optimum for both $ L(f(X^{\text{train}}, \Phi, \Theta)) $ and $ L(f(X^{\text{valid}}, \Phi, \Theta)) $. As long as this unique global optimum $(\Phi^\star, \Theta^\star)$ is the same for both functions and both functions behave properly, satisfying certain qualification assumptions as in [1] or [2], the convergence guarantee remains valid.
>
>
> This assumption is reasonable, as it implies that for both the training and target distributions, there exists an optimal set of expert and routing parameters for language modeling.  We plan to include this guarantee in the next iteration of our work to strengthen our results, and thanks for pointing this out.
>
>
>
> W3 & W4: Thanks for pointing it out. We will make sure that a brief introduction of the building blocks as well as references to the mentioned related works are included in the next iteration.
>
>
> W5: Yes, you are right about the privacy concerns. However, we would like to point out further that the only parameters we share are the global LoRA experts from the MLP layers, which is 38% of the total trainable parameters in the one-generalist-one-specialist case. With more specialists within each client, the ratio is even lower. Additionally, noise can be added to the global parameters to further guarantee differential privacy. We will add a more cautious and thorough discussion to the write-up.
>
>
> Q1: Our extra memory and computational complexity only come from the router, which is a small one-layer MLP. The size is 768*2 in our experiments. We refer to the second point of our **public comment** for a detailed analytical comparison.
>
>
> Q2: The absence of a theoretical analysis stems from our primary focus on the LLM community, where providing formal mathematical proofs is not a common convention. However, we agree that having a convergence guarantee would better justify our method and plan to include one in the future iteration of the manuscript.
>
>
> Q3: No, the privacy leakage does not scale with device capacity, as we earlier pointed out in W5, no matter how many experts one device has, the only shared part is the global experts, which in our settings is the very first expert from each MLP layer. Having more capabilities does not make one device more exposed to privacy leakage.
>
>
> [1] Alternating Minimizations Converge to Second-Order Optimal Solutions. Qiuwei Li, Zhihui Zhu, Gongguo Tang Proceedings of the 36th International Conference on Machine Learning, PMLR 97:3935-3943, 2019.
>
> [2] On the rate of convergence of alternating minimization for non-smooth non-strongly convex optimization in Banach spaces. Jakub Wiktor Both, Optimization Letters 16.2 (2022)

---

> ### Author Response · Authors · 2024-11-25
>
> Dear Reviewer HA1D,
>
> Thank you for the time and effort you have dedicated to reviewing our manuscript and providing thoughtful and positive feedback.
>
> Following your suggestions, we added a convergence result to the manuscript (see Theorem 3.1 from the modified manuscript). Moreover, we have clarified the communication and computational overhead as you pointed out in the global response, and added a new experiment to strengthen our empirical results. All of these changes can as well be found in the modified manuscript (highlighted in blue).
>
> We hope these adequately address your concerns and kindly ask for a re-evaluation of our work. We understand it is a particularly busy period for you, and we greatly appreciate your time in advance.
>
> Thank you,
>
> Submission 3571 Authors

---

> > ### Comment · Reviewer_HA1D · 2024-11-26
> >
> > Thank you for your comprehensive response and the clarifications you offered. After careful consideration, taking into account the comments from other reviewers, I have decided to retain my score.

---

> > > ### Author Response · Authors · 2024-11-26
> > >
> > > Dear Reviewer HA1D,
> > >
> > > Thanks for your response and your time invested in the re-evaluation of our work. Could you please let us know why you decided to maintain your score? If there are still major concerns from your side, we would be glad to offer further clarifications and supporting materials. Thank you for helping us improve our work.
> > >
> > > Best,
> > >
> > > Submission 3571 Authors

---

### Author Response · Authors · 2024-12-03

Dear Reviewers,

Thank you again for your feedback. As the extended discussion period comes to a close, we kindly request your consideration of our latest clarifications to ensure that the addressed concerns are accurately reflected in your score adjustments.

Thank you and best regards,

Submission 3571 authors

---

### Meta-Review · Area_Chair_pQHT · 2024-12-21

**Metareview:**

The paper presents a new method for building on-device language modeling. The proposed approach seeks to provide each person with a personalized language model using ideas from mixtures and federated learning. The personalized model on each device is a mixture -- built by sharing tokens across shared models (generalists) and local models (specialists). The routing mechanism ('i.e., which tokens to share') is determined by solving a bi-level optimization problem that uses data from the target domain to account for distribution shifts. The paper benchmarks the performance of the proposed approach for salient tasks on three real-world datasets that exhibit heterogeneity.

**Feedback from Reviewers:** This paper received mixed feedback from four reviewers. On the one hand, reviewers recognized the broader need for work in this area and appreciated the technical machinery. On the other hand, they highlighted two overarching concerns: (W1) computation overhead resulting from the use of bi-level optimization; (W2) validity of experimental claims (e.g., what is driving the gain, will it scale beyond GPT-2). Given the mixed feedback, I spent more time on this paper than usual -- reading through the reviews, the author responses, and the final submission. With regards to the weaknesses identified by reviewers:

- W1: I do not share the reviewers' concerns. My view is that research papers that introduce new methods should develop methods that offer new functionality or that improve functionality in a meaningful way. In such cases, the bar for "compute" should be that it is viable -- i.e. we should treat it as a constraint rather than an end in-and-of itself. It does not make sense to view compute as an objective since: (1) the new functionality may be worth the compute; (2) there are more effective avenues to reduce compute (e.g., an efficient implementation at production). Given this, the burden for methods papers is to show viability and functionality so that the method can be adopted and implemented efficiently in practice.

- W2: I do not share the reviewers concerns, but have different concerns. One of the reviewer's concerns in this case was whether the method would exhibit the same kinds of performance gains when using new generations of LLMs. During the rebuttal, this concern led the authors to run the experiments with LLAMA-3.2 and to report that the gains reported in their paper had disappeared for 2/3 datasets. In this case, I agree with the authors that this can be explained by test set contamination. However, the broader issue is that we have ended up in a position where we can no longer draw conclusions.

**Strengths and Weaknesses** Overall, I find that there is much to like about this paper. It targets a problem that is of broad interest (on-device learning), proposes an interesting method (federated learning with mixtures), and includes experiments that demonstrate viability and soundness. In my view, its main weakness is a failure to demonstrate significance. As it stands, the main "sell" for the method is that it can improve local performance by adapting to local datasets. This sell is not compelling because we expect such improvements when learning on-device models. On my end, it is also concerning since we have a limited ability to test such claims going forward.

In this case, I see a major opportunity to address this issue by drawing the focus back to mixtures. Given that their method is meant to learn a mixture, we should expect that these models will exhibit the same kinds of strengths as mixtures of LLMs (see e.g., [Wang et al.](https://arxiv.org/pdf/2406.04692)). I would use such papers as a starting point to identify "strengths" that can lead to meaningful improvements in functionality in the federated regime. These "strengths" could provide the basis for more compelling demonstrations in Section 4. As it stands, the experiments in this part are getting in the way of a compelling story. Specifically, the routing analysis in 4.2 simply highlights the behavior of the algorithm, and the gains observed under heterogeneity are somewhat expected.

**Recommendation**: I am recommending rejection at this stage. I want to be clear that this was a borderline case. Having read the paper, I find that the work is well-written and well-executed, but has simply been positioned in a way that will limit its impact. In this case, I ultimately believe that this work will end up as a far stronger contribution to the field following another round of revisions.

**Additional Comments On Reviewer Discussion:**

See above.

---

### Decision · Program_Chairs · 2025-01-22

Reject